# Modelling coarse and giant desert dust particles

Eleni Drakaki[1,2], Vassilis Amiridis[1], Alexandra Tsekeri[1], Antonis Gkikas[1], Emmanouil Proestakis[1], Sotirios Mallios[1], Stavros Solomos[3], Christos Spyrou[1], Eleni Marinou[1,4], Claire L. Ryder[5], Demetri Bouris[6], and Petros Katsafados[2]

[1]IAASARS, National Observatory of Athens, Athens GR-15236, Greece
[2]Harokopion University of Athens (HUA), Department of Geography, Athens GR-17671, Greece
[3]Academy of Athens, Research Centre for Atmospheric Physics and Climatology, Athens GR-10679, Greece
[4]Institut für Physik der Atmosphäre, Deutsches Zentrum für Luft- und Raumfahrt, Oberpfaffenhofen, Germany
[5]University of Reading, Department of Meteorology, Reading, RG6 6BB, UK
[6]National Technical University of Athens, School of Mechanical Engineering, Athens, GR-15780, Greece

*Correspondence to*: Eleni Drakaki (eldrakaki@noa.gr)

**Abstract.** Dust particles larger than 20 μm in diameter have been regularly observed to remain airborne during long-range transport. In this work, we modify the parameterization of the mineral dust cycle in the GOCART-AFWA dust scheme of WRFV4.2.1, to include also such coarse and giant particles, and we further discuss the underlying misrepresented physical mechanisms which hamper the model in reproducing adequately the transport of the coarse and giant mineral particles. The initial particle size distribution is constrained by observations over desert dust sources. Furthermore, the Stokes' drag coefficient has been updated to account realistic dust particles sizes (Re $< 10^5$). The new code was applied to simulate dust transport over Cape Verde in August 2015 (AER-D campaign). Model results are evaluated against airborne dust measurements and the CALIPSO-LIVAS pure dust product. The results show that the modelled lifetimes of the coarser particles are shorter than those observed. Several sensitivity runs are performed by reducing artificially the particles' settling velocities in order to compensate underrepresented mechanisms, such as the non-spherical aerodynamics, in the relevant parameterization schemes. Our simulations reveal that particles with diameters of 5.5-17 μm and 40-100 μm are better represented under the assumption of a 80% reduction in the settling velocity (UR80) while particles with sizes ranging between 17μm and 40 μm are better represented in a 60% reduction in settling velocity (UR60) scenario. The overall statistical analysis indicates that the best agreement with airborne in-situ measurements downwind (Cape Verde) is achieved with a 40% reduction in settling velocity (UR40). Moreover, the UR80 experiment improves the representation of the vertical structure of the dust layers as those are captured by the CALIPSO-LIVAS vertically-resolved pure dust observations. The current study highlights

the necessity of upgrading the existing model parameterization schemes of the dust life-cycle components towards improving
the assessment of the dust-related impacts within the Earth-Atmosphere system.

## 1 Introduction

Dust is the most prominent contributor to the global aerosol burden, in terms of dry mass, and it ranks second in
aerosol emissions (Gliß et al., 2021; Huneeus et al., 2011; Textor et al., 2006). The major sources of dust are situated across
the "dust belt" (Prospero et al., 2002) stretching in the Northern Hemisphere hosting deserts and erodible soils (e.g.,Goudie
and Middleton, 2006), that are prone to windblown dust. Most of the global dust budget comes from the Sahara Desert,
followed by deserts in the Middle East and Asia. (Ginoux et al., 2012; Huneeus et al., 2011; Kok et al., 2021; Li and Osada,
2007). Spatially more limited desert regions in the Southern Hemisphere emit lower amounts of mineral particulate matter
(Ginoux et al., 2012; Huneeus et al., 2011; Kok et al., 2021; Li and Osada, 2007), and less than 5% comes from high-latitude
sources (Bullard et al., 2016).
Dust particles act as ice nuclei (IN) on cold cloud processes (Marinou et al., 2019; Solomos et al., 2011) and when
mixed or coated with hygroscopic material, they can affect warm cloud processes (Twohy et al., 2009) and serve as cloud
condensation nuclei (CCN). Dust particles rich in key micronutrients such as iron (Fe) and phosphorus (P) affect
biogeochemical processes in marine and terrestrial ecosystems (Jickells et al., 2005; Okin et al., 2004; Stockdale et al., 2016;
Tagliabue et al., 2017) and disrupt the carbon cycle (Jickells et al., 2014) after their wet and dry deposition. Severe dust
episodes can affect aviation and telecommunications (Harb et al., 2013; Weinzierl et al., 2012; Nickovic et al., 2021), human
health (e.g., Du et al., 2016; Giannadaki et al., 2014) and solar energy production (Kosmopoulos et al., 2018).
Apart from the dust load intensity, the size of the suspended mineral particles plays a determinant role on the related
impacts on weather and climate, among others. Larger dust particles act more efficiently as CCN (Petters and Kreidenweis,
2013) and IN (Diehl et al., 2014) altering cloud microphysical processes and subsequently the hydrological cycle. Recent
studies suggest that coarser dust aerosols are more effective absorbers of the incoming solar radiation, enhancing atmospheric
warming (Mahowald et al., 2014; Ryder et al., 2019). Therefore, it is imperative to represent realistically the dust particle size
distribution (PSD) facilitating a thorough investigation on the dust transport processes and the dust-induced impacts.
Airborne dust particles has been observed to sizes up to 300 μm, whereas even larger particles with diameters up to
450 μm have been recorded from in situ deposition measurements acquired at buoys mounted across the Tropical Atlantic
Ocean (van der Does et al., 2018). Dust particles are usually divided into three different modes (fine, coarse and giant) without
strictly defined bounds of their sizes (Goudie, 2014; Knippertz and Stuut, 2014). According to Ryder et al. (2019), the fine
mode represents dust particles with $D \leq 2$ μm, the coarse mode those with diameters between 2 μm  and 20 μm, and the giant
mode particles with $D \geq 20$ μm. A recent study (Ryder et al., in preparation) suggests that the above modes can be further
descritized into four categories, namely fine (D < 2.5 μm), coarse (2.5-10 μm), super-coarse (10-62.5 μm), and giant (D > 62.5
μm).

The existence of dust particles larger than 20 μm in diameter was already demonstrated in the 1970s based on measurements in the Caribbean (Prospero et al., 1970). Nevertheless, these sizes were neglected in atmospheric dust models since giant particles were assumed to be rare. This assumption has been disproved in recent decades by a large number of airborne campaigns equipped with state-of-the-art in situ and remote sensing instruments. Specifically, in the framework of the SAMUM1 (Weinzierl et al., 2009) and SAMUM2 (Liu et al., 2018) experimental campaigns it has been justified that above sources dust aerosols up to 40 μm in diameter were recorded in 20% of the identified dust layers, while over Cape Verde mineral particles up to 30 μm in diameter were measured (Weinzierl et al., 2011). This reduction of dust particle sizes, along the transport pathway, is attributed to the gravitational settling. Similar findings were reported in the FENNEC campaign (Ryder et al., 2013b) with mean effective particle diameters ranges of 22 to 28 μm and 15 to 18 μm for fresh and aged dust, respectively. During the AER-D campaign, in the Saharan outflow zone near Cape Verde and the Canary Islands, mineral particles with diameters larger than 20 μm were systematically recorded, while in 36% of the studied cases, particles with diameters larger than 40 μm recorded (Ryder et al., 2018). Dust particles with diameters of 10 to 30 μm were detected during the SALTRACE campaign in Barbados (Weinzierl et al., 2017), revealing that they were suspended far away from their sources at about 2000 km more than what would be expected from the Stokes' theory (Weinzierl et al., 2017). Atmospheric dust models are the optimal tool to simulate the components of the dust cycle and therefore to study the dust-related effects. However, the state-of-the-art atmospheric dust models are characterized by inherent limitations in accounting for realistic emission and transport dust size distributions (Huang et al., 2020; Kok, 2010; Mahowald et al., 2014). To overcome these model drawbacks, it is needed to extend the PSD towards the giant particles size spectrum in order to shed light on the processes that sustain larger dust aerosols in the atmosphere for longer periods than expected.

Ginoux, (2003) modeled dust aerosols up to 70 μm in diameter using the Global Ozone Chemistry Aerosol Radiation and Transport (GOCART) model and examined the effects of non-sphericity assuming randomly oriented ellipsoidal particles. His results showed that the reduction of the settling velocity results in a better agreement with observations when the aspect ratio is equal or greater than 5. The new modeled particle size distributions (PSDs) were in generally better agreement with the AERONET observations, although the PSDs were significantly underestimated for diameters near 10 μm. The aspect ratio of 5 results in a reduction in settling velocity of about 45% for particles with sphere volume-equivalent diameters near 10 μm and 60% for particles with sphere-volume-equivalent diameters near 30 μm. Maring et al. (2003) applied a simple empirical model and suggested that an upward velocity of 0.0033 ms$^{-1}$ (0.33 cm s$^{-1}$) is required to accurately predict PSD changes during transport. Although their comparisons were limited to sizes up to 25 μm, they pointed out that unknown or not well-known processes counteract gravity settling. Possible mechanisms which can interpret the aforementioned findings are (i) vertical mixing within the Saharan air layer during the day (Gasteiger et al., 2017), (ii) the lower settling velocities of non-spherical dust particles (Huang et al., 2020; Mallios et al., 2022), (iii) the underrepresented meteorological conditions (O'Sullivan et al., 2020), (iv) the unresolved turbulence (Gu et al., 2021), (v) the electrification of dust (Daskalopoulou et al., 2021; Mallios et al., 2021a; Mallios et al., 2022; Joseph R. Toth III et al., 2020; Renard et al., 2018; Nicoll et al., 2011) and (vi) the numerical errors that perturb the mass balance (Ginoux, 2003).

In this work, we demonstrate for the first time a method for incorporating coarse and giant desert dust particles (D >
20 μm, following the definition of the dust modes proposed in Ryder et. al, (2019), into the Advanced Research Weather
version of the Weather Research and Forecasting (WRF-ARW) model in conjunction with the GOCART (Ginoux, 2003)
aerosol model and the Air Force Weather Agency (AFWA) dust emission scheme (LeGrand et al., 2019) (WRF-GOCART-
AFWA model). After pinpointing that the model quickly deposits coarse and giant dust particles, we investigate the reasons
behind those findings: We use sophisticated in situ PSD measurements to initialize the model over the sources and to evaluate
the simulated PSD over the receptor areas. We also use pure-dust spaceborne retrievals to assess the model performance in
terms of reproducing the vertical structure of the dust layers. In addition, we perform a series of sensitivity tests by reducing
the settling velocity of mineral particles in the model and we investigate the concomitant effects on dust fields. The article is
organized as follows: In Sect. 2, we describe the methodology in terms of the changes we made to the code of WRF-GOCART-
AFWA, the setup of the model and the experiments performed, and the observational data we used for model validation. The
results of our work are presented in Sect. 3, Sect. 4 contains the discussion and Sect.5 the summary and the conclusions of this
work.

## 2 Model and Data

### 2.1. WRF-GOCART-AFWA model

In our numerical experiments to study the transport of coarse and giant dust aerosols, we use the WRF-ARWv4.2.1
model coupled with the GOCART aerosol model and the AFWA dust emission scheme (LeGrand et al., 2019). The current
version of the WRF-GOCART-AFWA model accounts for giant dust particles in the calculated dust emission fluxes (up to
125 μm) and assumes that the transported dust particles are up to 20 μm in diameter. To extend the transport PSD to coarser
and giant mineral particles, we implemented several developments in the standard WRF-GOCART-AFWA model, which are
discussed in Sect. 2.1.1 and 2.1.2. Figure 1 shows a sketch of the workflow: the first three steps refer to the implemented
modifications in the standard WRF-GOCART-AFWA code: In **step 1**, we establish the definition of a prescribed PSD for the
emitted dust particles at the source based on in situ reference measurements, and we distribute the total emitted dust
accordingly; in **step 2**, we define five size ranges (five model size bins) for the transported PSD covering dust particle sizes
(in diameter) spanning from 0.2 μm to 100 μm (Sect. 2.1.1); in **step 3**, we implement an updated drag coefficient that applies
to the sizes of the entire range of Aeolian dust PSD (Sect. 2.1.2). These code upgrades are integrated into the new WRF-L
model. Table 1 shows the properties of the size bins in the standard WRF-GOCART-AFWA code and the size classes defined
in the new WRF-L code. At **step 4** we perform model experiments and validate the model results using different model
configurations against observations (Sect. 2.2), as described in detail in Sect. 3.

### 2.1.1 Dust size distribution

In observational studies of non-spherical particles, it is customary to describe their size in terms of spherical volume equivalent diameter. Here, to describe particles' sizes distributed within the five size bins of the WRF-L model, we use the sphere-volume-equivalent effective diameter ($D_{eff}$), which is more relevant to the optical properties of the particles (Hansen and Travis, 1974). In this way, we simplify the comparison between the model calculations and the observations of the optical properties of the particles (e.g., dust optical depth). The $D_{eff}$ in (μm) of each size bin is calculated as shown in Eq. 1, and is provided in Table 1.

$$D_{eff} = \frac{\int_{D_{lo,k}}^{D_{u,k}} D^3 \cdot \frac{dN}{dD} \cdot dD}{\int_{D_{lo,k}}^{D_{u,k}} D^2 \cdot \frac{dN}{dD} \cdot dD},$$

(1)

Where $D$ is the particle diameter in (m) and $\frac{dN}{dD}$ is the particle number size distribution in number of particles per m$^3$. The parameters at each size bin $k$ are listed in Table 1. Henceforward, references about the size of the particle correspond to particle volume equivalent effective diameter, unless mentioned otherwise.

In the default GOCART-AFWA dust emission scheme of WRF, the total emitted vertical dust flux is estimated at each grid point prone to dust emission, when favorable conditions are met. The dust flux is then distributed over five transport size bins, based on the fragmentation theory of Kok, (2011), bounded to diameters up to 20 μm. Since our goal is to consider larger dust particles than those commonly used in the current atmospheric dust models, we redefine the five transport model bins including particles with diameters up to 100 μm (Table 1). We rely on prescribed PSD for the emitted dust particles at the source based on the airborne in situ measurements acquired during the FENNEC campaign of 2011 (Ryder et al., 2013a). More specifically, for the freshly uplifted dust we use the mean PSD at the lowest available height (i.e., 1km), obtained by averaging profile measurements above the Sahara (Mauritania and Mali), hereafter called the "observed FENNEC-PSD", which is shown in Fig. 2(a) with red squares. Figure 2a shows also the "fitted FENNEC-PSD" (solid red line), which is the fit of the "observed FENNEC-PSD", using five lognormal modes (Table 4). In Sect. 2.2.1 more information is provided on the derivation of the mean "observed FENNEC-PSD", including also the description of the FENNEC 2011 campaign, the in-situ instrumentation used and the processing of the acquired data. Based on the FENNEC-PSD we calculate the mass fraction ($k_{factors}$) distributed among the redefined transport model size bins in Eq. 2. The weighting factors $k_{factors}$ are shown in Fig.2(b).

$$k_{factors} = \frac{\int_{Dlo,k}^{Du,k} \frac{1}{D} \frac{dV}{dlnD} \cdot dD}{\int_{Dlo,k_{min}}^{Du,k_{max}} \frac{1}{D} \frac{dV}{dlnD} \cdot dD},$$

(2)

Where $D$ is the particle diameter, $\frac{dV}{d\ln D}$ is the volume size distribution in $\mu m^3 cm^{-3}$, $D_{lo,k}$ and $D_{u,k}$ are the margins of each size
bin $k$ in $\mu m$.

### 2.1.2 Updated gravitational scheme

In the GOCART-AFWA dust scheme of WRF, the forces acting on a dust particle moving along the vertical direction

are the gravitational force $F_g$ and the aerodynamic drag force $F_{drag}$, which are mathematically expressed in Eq.3 and Eq.4,
respectively.

$$F_g = \rho_p \cdot V_p \cdot g, \tag{3}$$

$$F_{drag} = \frac{1}{2} \cdot \frac{C_D}{C_{cun}} \cdot A_p \cdot \rho_{air} \cdot u_{term}^2, \tag{4}$$

Where $\rho_p$ stands for particle density in $kgm^{-3}$, g corresponds to the gravitational acceleration in $ms^{-2}$, $V_p = \frac{1}{6} \cdot \pi \cdot D_{eff}^3$ is the
particle volume in $m^3$ and $A_p = \frac{\pi}{4} \cdot D_{eff}^2$, is the particle's projected area normal to the flow in $m^2$, $\rho_{air}$ is the air density in
$kgm^{-3}$. and $D_{eff}$ represents the particles' diameter in $m$ for each model size bin (assuming spherical particles, as defined in
Sect. 2.1.1). $C_D$ is the aerodynamic drag coefficient (unit less) and $C_{cun}$ is the slip correction to account for slip boundary
conditions (Davies, 1945) and it is expressed as a function of the air mean free path ($\lambda$, in meters) (Eq. 5):

$$C_{cun} = C_{cun}(\lambda) = 1.0 + \frac{2 \cdot \lambda}{D_{eff}} [1.257 + 0.4 \cdot e^{\frac{-1.1 \cdot D_{eff}}{2 \cdot \lambda}}], \tag{5}$$

The constant velocity that a particle builds up falling vertically within the Earth's atmosphere, is defined as the terminal settling
velocity $u_{term}$, and it can be estimated by solving the 1-D equation of motion at the steady state limit, where net force is
assumed to be equal to zero:

$$\rho_p \cdot V_p.g = \frac{1}{2} \cdot \frac{C_D}{C_{cun}} \cdot A_p \cdot \rho_{air} \cdot u_{term}^2, \tag{6}$$

In the default GOCART-AFWA dust scheme the drag coefficient is given by Stokes' Law and is defined as:

$$C_D = \frac{12}{Re}, \tag{7}$$

Where $Re$ is the Reynold's number (unit less) given by the following equation as a function of the particle volume equivalent
effective diameter $D_{eff}$:

$$Re = \frac{\rho_{air} \cdot u_{term} \cdot D_{eff}}{2 \cdot \mu}, \tag{8}$$

Where $\mu$ is the air dynamic viscosity in $\frac{kg}{m \cdot s}$ defined as a function of air temperature $T$ in $K$ by the following equation (Hilsenrath,
1955; United States Committee on Extension to the Standard Atmosphere., 1976):

$$\mu = \frac{\beta \cdot T^{\frac{3}{2}}}{T+S}, \tag{9}$$

where $S$ is the Sutherland constant which equal to $110.4\ K$ and $\beta$ is a constant which equals to $1.458 \cdot 10^{-6}\ kg \cdot m^{-1} \cdot s^{-1} \cdot$
$K^{-1/2}$ .
and the air mean free path is expressed as:
$$\lambda = \frac{1.1 \cdot 10^{-3} \cdot \sqrt{T}}{P} \tag{10}$$
Where $T$ is the air temperature in $K$ and $P$ the air pressure in $hPa$.

The slip-corrected drag coefficient of the Stokes' Law ($\frac{12}{Re \cdot C_{cun}}$) is valid only for Re <<1, thus it is not representative
for particles with $D_{eff}$ larger than ~10 μm. Therefore, an adaptation of the drag coefficient is needed in order to be valid for
higher Re values (i.e., 0<Re<16), since in our work dust particles with diameters larger than 20 μm are considered. To realize,
we use the drag coefficient $C'_D$ (Eq. 11), proposed by Clift and Gauvin, (1971):

$$C'_D = \frac{12}{Re} \cdot (1 + 0.2415 \cdot Re^{0.687}) + \frac{0.42}{1+\frac{19019}{Re^{1.16}}}, \ for \ \ Re < 10^5 \tag{11}$$

Mallios et al., (2020) used the same $C'_D$ as a reference for the development of a drag coefficient for prolate ellipsoids, as more
suitable for $Re < 10^5$. The departures between the drag coefficients given by Stokes and Clift and Gauvin (1971) become
more evident for increasing particles' sizes. More specifically, the drag coefficient given by Clift and Gauvin (1971) can be
up to 2 times higher than those of the Stokes' Law for coarse and giant particles (Fig. S1).
In the default WRF code the slip correction is applied unconditionally for all the Re values, probably without affecting
the solution significantly due to the small particle sizes ($D_{eff} < 20\ \mu m$). However, in our work a condition is required for
applying the slip correction only in the Stokes' regime (e.g. Re < 0.1, Mallios et. al, 2020). Hence, we apply the bisection
method to calculate the terminal velocity for each model size bin using the revised drag coefficient and, at first, ignoring the
slip correction. When the solution lies in the Stokes' regime (e.g. Re < 0.1), we recalculate the settling velocity using the
corrected drag coefficient $C'_{D,slip} = \frac{C'_D}{C'_{cun}}$, where $C'_{cun} = C_{cun}(\lambda')$ with $\lambda'$ the mean free path obtained by (Jennings, 1988):

$$\lambda' = \sqrt{\frac{\pi}{8}} \cdot \frac{\frac{\mu}{0.4987445}}{\sqrt{P\rho_{air}}}, \tag{12}$$

### 2.1.3 Model experiments

Using the WRF-L code, we first run the CONTROL experiment. Our simulation period coincides with the AER-D

experimental campaign (29/7 - 25/8/2015) for a domain bounded between the 1.42°N and 39.99°N parallels and stretching
between the 30.87°W and 46.87°E meridians (Fig. 3). The simulation area encompasses the major Saharan also including the
downwind areas in the eastern Tropical Atlantic. We use an equal-distance grid with a spatial grid spacing of 15 km x 15 km
consisting of 550 × 300 points whereas in vertical, 70 vertical sigma pressure levels up to 50 hPa are utilized (defined by the
model). The simulation period consists of nine 84-hour forecast runs, which are initialized at 12 UTC, using the 6-hour Global
Forecast System Final Analysis (GFS - FNL) reanalysis product, available at a 0.25°x0.25° spatial grid spacing. The sea surface
temperatures, acquired by the NCEP daily global SST analysis (RTG_SST_HR), are updated every six hours along with the
lateral boundary conditions. Topography is interpolated from the 30-sec Global Multi-resolution Terrain Elevation Data 2010
(GMTED2010, Danielson and Gesch, (2011)). Land use is defined based on the Moderate-resolution Imaging
Spectroradiometer (MODIS) observational data, modified by the University of Boston (Gilliam and Pleim, 2010). From each
84-hours cycle, the first 12 hours are discarded due to model spin up. Likewise, the first week of the simulation served as a
spin-up run for the accumulation of the background dust loading and it is excluded from the analysis. The simulation runs are
performed in a dust-only mode, neglecting the radiative feedback from aerosols. We scale the dust source strength, by tuning
the empirical proportionality constant in the horizontal saltation flux equation (in eq. 10 in LeGrand et al., (2019)) in order to
obtain the best match between the modeled DOD and the AERONET AOD (RMSE=0.34, bias=-0.07) acquired at 8 desert
stations: Banizoumbou, Dakar, El_Farafra, Medenine- IRA, Oujda, Tizi_Ouzou, Tunis_Carthage, Ben_Salem). Note that we
take into account only AERONET records when AODs are higher than 0.2 (Version 3.0, Level 1.5, Giles et al., 2019; Sinyuk
et al., 2020) and the Angstrom exponent is lower than 0.75. The tuning constant is equal to 3 and is applied throughout the
model domain. The complete configuration options for the run are listed in Table 2. The resolution applied in this study (15km
grid spacing) is adequate for the scale of phenomena we want to study, improves the representation of topography and increases
the accuracy of the reproduced weather and dust fields compare to coarser resolution such as used in global datasets (e.g. 0.5
deg GFS) (Cowie et al., 2015; Basart et al., 2016; Roberts et al., 2017; Solomos et al., 2018). WRF-Chem solver uses a 5th-
order horizontal advection scheme and a 3-rd order vertical advection scheme to solve the scalar conservation equation, along
with the 3-rd order Runge-Kutta time integration scheme (Grell et al., 2005). The use of such high-order advective schemes
eliminate the numerical errors of diffusion in the code. We should note though that in the deposition parameterization of

GOCART-AFWA dust scheme the vertical advection of the losses due to the gravitational settling is solved by a first order explicit scheme, which is notoriously too diffusive (Versteeg and Malalasekera, 2007) and thus it can possibly induce numerical errors in the mass conservation (Ginoux, 2003). A series of additional sensitivity runs has been performed aiming to resemble possible mechanisms (misrepresented or even absent in the model) counteracting gravitational settling towards reducing the differences between the CONTROL run calculations and the in-situ observations (shown in Sect. 3.4). To be more specific, we gradually reduced (with an incremental step of 20%) the settling velocity by up to 80%, with the corresponding runs named as URx (x corresponds to the reduction in percentage terms). Under such theoretical conditions, it is expected that the giant dust particles will be suspended for longer periods and that they will be transported at larger distances than the current state-of-the-art models simulate, failing to reproduce what is observed in the real world. Based on these sensitivity experiments, we defined a constant (by percentage) relevant reduction of the particles' settling, which in its absolute value varies with size. Therefore, it is more similar to the effects that are related to aerodynamic forces due to the non-spherical shape and the orientation of the suspended dust particles (Ginoux, 2003; Loth, 2008; Zastawny et al., 2012; Shao et al., 2017; Sanjeevi et al., 2018; Mallios et al., 2020). Finally, the full list of the performed experiments is given in Table 3.

**2.1.4 Dust extinction coefficient**

For the evaluation of the model mid-visible (550 nm) dust extinction profiles the corresponding products from the Lidar climatology of Vertical Aerosol Structure for space-based lidar simulation studies (LIVAS) dataset is used as reference. For the spatiotemporal matching between the modelled and the observed dust extinction, we first project the two datasets onto a common horizontal grid, by converting the model outputs from their native horizontal grid spacing (15 km x 15 km) to the structured 1°x1° equal lat-lon grid of LIVAS. The model extinction coefficient for each size bin $k$ ($EC_{550,k,n,l}$) is then calculated at each grid cell $n$ and within each model level $l$, as shown in Eq.13.

$$EC_{550,k,l,n} = \sum_1^k \frac{3}{2\rho_{,k}D_{eff,k}} M_{n,k,l} Q_{ext550,k}, \tag{13}$$

where $M_{n,k,l}, \rho_k, D_{eff,k}$ and $Q_{ext550,k}$ are the grid cell dust mass concentration in g/m³, the particle density in g/m³, the effective diameter in m, and the extinction efficiency factor at 550 nm, of size bin k.

$Q_{ext550,k}$ is calculated using the Mie scattering code (Mie, 1908), considering spherical dust particles, and a refractive index of 1.55 + i0.005, which is representative of dust (e.g. Dubovik et al., 2002). Although the extinction coefficient values for spherical particles may be different from the extinction coefficient values of the dust particles, which have irregular shapes, to our knowledge there is no data available for the extinction coefficient of the latter. The extinction coefficient values of spheroidal shapes, commonly used as a proxy of the dust shapes, are not substantially different compared to the spherical particles (Tsekeri et al., 2022), at least when considering the aspect ratios measured for dust particles in Sahara (Kandler et al., 2009). For simplifying the computations, we assume that the particles in each size bin have the same size (i.e. $D_{eff,k}$), and thus

the same $Q_{ext550,k}$. In vertical, the fine resolution LIVAS dust extinction coefficient is rescaled (averaging) to match the model
layers vertical margins. In the time dimension, the model outputs at the closest lead times to the satellite overpass are selected.

## 2.2 Observational datasets

### 2.2.1 Airborne in situ observations

During the FENNEC field campaign in 2011 (Ryder et al., 2013b, 2013a) and the AER-D field campaign in 2015

(Ryder et al., 2018, 2019), airborne in situ observations were collected with the FAAM BAE research aircraft. In this study
we use size distributions from the FENNEC field campaign, aquired during aircraft profiles over the Sahara (Mauritania and
Mali), as described in Ryder et al. (2013a). We select size distributions from "freshly uplifted dust" cases, when dust particles
are in the atmosphere for less than 12 h. Additionally, from these profiles we use data from the lowest available altitude,
centered at 1km, covering altitudes between 0.75 to 1.25km. The derived PSD is depicted in Fig.2(a), hereafter referred to as
the "observed FENNEC-PSD". Error bars in Fig.2(a) indicate the standard deviation of the observed values across the profiles
and altitudes we used. The instrumentation for those measurements was the Passive Cavity Aerosol Spectrometer Probe
(PCASP, 0.13-3.5 μm), the Cloud Droplet Probe (CDP, 2.9-44.6 μm), using light scattering measurements and assuming a
refractive index (RI) of 1.53-0.001i (which is constant with particle size), spherical shape for the particles, and using Mie
calculations to convert from optical to geometric diameter, as well as the Cloud Imaging Probe (CIP15, 37.5-300 μm)). The
instruments and data processing are described in Ryder et al. (2013a). The midpoint size bin diameters do not overlap, though
there is some overlap in bin edges between the instruments. A fit on the observations is provided in Figure 2a (the "fitted
FENNEC-PSD" with solid red line), which is used in the parameterization of the emitted dust, as described in Section 2.1.1,
to modify the GOCART-AFWA dust scheme in WRF.

We also use PSD observations during horizontal flight legs at a constant height (referred either as RUNs or flight

segments) over the Atlantic Ocean during AER-D. We use measurements taken with PCASP (D =0.12-3.02 μm) for fine dust
particles. For the coarse and giant mode of dust we used measurements from CDP (D=3.4-20 μm, although CDP measurements
availability extends up to 95.5 μm as it is explained below) and the two-dimension Stereo probe (2DS, D = 10–100 μm -
although the instrument measures up to 1280 μm few particles larger than 100μm were detected). For the light scattering
techniques of PCASP and CDP, a RI = 1.53-0.001i is assumed for the conversion of the optical to geometric diameter (as in
FENNEC 2011 campaign). CDP observations extend up to the size of 95.5 μm, thus data from CDP and 2DS partly overlap in
their size range. Since 2DS observations are more reliable in the overlapping size range, we used the CDP observations for
particles with sizes up to 20 μm. Also, 2DS-XY observations are preferred over the 2DS-CC, since they better represent the
non-spherical particles. A more detailed description of the in-situ instruments and the corresponding processing of the data
acquired during the AER-D campaign is included in Ryder et al., (2018). The error bars represent the total (random and
systematic) measurement error due to the counting error, the discretization error, the uncertainties in the sample area and the
uncertainties in the bin size due to Mie singularites (Ryder et al., 2018). All PSD measurements are at ambient atmospheric
conditions. The locations of the flights of AER-D used in this study are depicted in Fig.3.

**2.2.2 LIVAS product**

For the validation of the vertical distribution of dust from the model (see Sect. 3.5), we utilize the pure-dust profiles
provided by the LIVAS dataset, originally presented in Amiridis et al. (2013; 2015) and updated in Marinou et al. (2017). The
LIVAS pure-dust product is a global dataset, covering the period between 06/2006 and 05/2020, and is provided a) on per-
granule level with similar resolution to the original Cloud-Aerosol Lidar and Infrared Pathfinder Satellite Observations
(CALIPSO) L2 profile products (i.e., 5 km horizontal and 60 m vertical), and b) as a global three-dimensional database of
monthly-mean averaged profiles of aerosol properties, on a uniform horizontal grid spacing of $1° × 1°$. LIVAS was developed
applying the dust-separation technique described in Tesche et al., (2009) on the CALIPSO level 2 version 4 products (Winker
et al., 2009). The LIVAS pure-dust product has been used to a variety of dust-oriented studies including the investigation of
the dust sources and the seasonal transition of the dust transport pathways (Marinou et al., 2017; Proestakis et al., 2018); the
evaluation of the performance of atmospheric and dust transport models (e.g. Tsikerdekis et al., 2017; Solomos et al. 2017;
Georgoulias et al., 2018; Konsta et al., 2018), the evaluation of new satellite-based products (e.g. Georgoulias et al., 2016;
Chimot at al. 2017; Georgoulias et al., 2020; Gkikas et al., 2021), and on dust assimilation experiments (Escribano et al., 2021).
Herein, the LIVAS pure-dust extinction product is used for the assessment of the simulated dust vertical patterns. In the
geographical region of our study, the uncertainty of the product is estimated to be less than 20% in altitudes up to 6km (Marinou
et al. 2017).

**3 Results**

**3.1 Settling Velocities**

Figure 4 shows the altitude profiles of the settling velocities for each size bin from the CONTROL run, averaged over
the simulation domain, and the period of interest. Settling velocity is increases for larger mineral particles. The terminal
velocities for particles within bin 5 are two orders of magnitude higher than those in bin 2 and bin 3, and one order of magnitude
with respect to bin 4. An altitude dependency, regulated by the thermodynamic state of the atmosphere, of the terminal
velocities is also apparent in Figure 4, showing that they increase with height due to the reduction either of temperature or air
density (Eqs. 9 and 13) For the CONTROL run the average settling velocities near the surface are lower by approximately
10% than those at 6 km height, and this non-negligible reduction can be critical, particularly for coarser and giant particles
where velocities are higher.

### 3.2 Dust above the sources

In Fig. 5 we present how the PSD varies with height above an emission point (latitude=24.9º and longitude=9.2º) in Mali, on 11/08/2015 at 14UTC. The model PSDs are only from that grid model box interpolated at 1, 2, and 3 km height and for the particular timestep (11/08/2015 at 14UTC). The red squares correspond to the "observed FENNEC-PSD" sorted into the five bins. The error bars provide the maximum and minimum limits of the "observed FENNEC-PSD", sorted into the five model size bins, after including the standard deviation of "observed FENNEC-PSD". The "observed FENNEC-PSD" (see Section 2.2.1) has been derived from several flights above dust sources, thus it is representative of the PSDs above Sahara sources and it used here as reference. The black squares depict the "fitted FENNEC-PSD" sorted into five bins, used in the model parameterization to calculate the emitted dust mass of the corresponding five model transport bins. The difference between the "fitted FENNEC-PSD" and the "fitted FENNEC-PSD" occurs due to the fitting process. The modelled volume concentration is reduced with height by an order of magnitude between 2 and 3 km for particles with diameters 17-40 μm (bin 4). At 3km the simulated concentrations of particles in bin 4 and bin 5 are very low compared to the measurements in Fig. S2a of Ryder et al., (2013a) which indicate the removal of giant particles above 4 km (Ryder et al., 2013a, Figure S2a). Although a direct comparison between the modelled and the observed PSD for this particular emission point is not feasible, since the FENNEC campaign took place on different dates than the AER-D and there are no available measurements above dust sources for the period we performed our simulations, we note a modification of the PSD shape, both for model and observations at 1km. It is evident that the model overestimates the PSD for bins 1-3 while the opposite is found in the size spectrum of the super-coarse (bin4) and giant (bin5) dust particles. Therefore, a model weakness is revealed at the very early phase of the dust transport. Those differences can be attributed to an overestimation of their loss during uplift from the surface to 1 km, or to higher updrafts that remain unresolved in our numerical experiment. Another possible source of this underestimation could be the utilization of a not well-defined PSD shape constraining the distribution of emitted dust mass to the model transport size bins. The use of a PSD with a higher contribution of coarse and giant dust particles could possibly improve the representation of the coarse and giant particles aloft (Fig. S2 and S3) and can be assessed in future studies. Additionally, comparing the "observed FENNEC-PSD" with the modelled PSD of the scenario with the maximum relative reduction of the settling velocities (UR80) in Fig. 5, we find a significant increase of the modelled volume concentrations, reducing the differences seen in volume concentrations in bin4 and bin5 without the reduction of the settling velocity, although the underestimation in bin 5 is still evident.

### 3.3 Mean dust load

In Fig. 6, the spatial patterns of the columnar dust concentrations are depicted, averaged over the period of 5/8/15-25/8/15, for the total mass as well as for each one of the five size bins simulated with the CONTROL run. Among the first three bins, there are evident many similarities of the dust load spatial features, with maximum values in the Western Sahara whereas the dust advection pathways towards the Atlantic Ocean are clearly seen. In terms of intensity, the mass increases

from bin1 to bin 3 (5.5 – 17 μm), yielding the maximum values throughout the size ranges. Dust particles with diameters
between 17 μm and 40 μm (bin 4) are found mainly over land, and are subjected to short-range transport westwards (i.e., off
the Moroccan coast). Giant particles (bin 5) are found at very low concentrations ($< 0.5$ gr m$^{-2}$), at isolated areas over/near dust
sources, since the strong impact of gravitational settling prohibits their accumulation and transport.

**3.4 Dust size distribution**

Figure 7 illustrates the simulated PSDs, from each experiment (i.e., CONTROL and URx), along with those acquired
by the airborne in situ measurements at different segments and altitudes of the flight b928 in the surrounding area of Cape
Verde (downwind region). For the other AER-D flights (i.e., b920, b924, b932 and b934) similar findings are drawn and for
brevity reasons are omitted here and are included in the supplementary material (Fig.S4). All AER-D measurements
demonstrate the impacts of the processes that are associated with dust transport. The red squares represent the observations
and the error bars represent the total (random and systematic) measurement error (see Sect 2.2.1). The modelled PSDs are
collocated in space and time with the measurements of each flight segment. For each flight segment, we extract the modeled
PSD by interpolating the dust field to the specific altitude of the flight RUN. Additionally, we average the dust field of the
nearest grid cell to each coordinate pair along the flight segment track, and the eight neighbouring grid cells of the same
altitude. The coordinates of the flight leg track are depicted with orange dots and the collocated grid points used for deriving
the modelled PSD (at the specific height of each flight leg) with blue dots. In the time dimension, we average the two hourly
model outputs that contain the times of the measurement. In case that the time of measurement coincides with the exact hourly
output, the model output on that hour along with the outputs prior and after that are averaged. The error bars in the model PSDs
indicate the standard deviation of the collocated grid points averaging in space and time.
Based on our findings, for the CONTROL run, the model performs considerably well particularly near the surface
and above 4 km, reproducing the volume concentration of the particles residing within bins 1 and 2. Underestimations are
found for the third bin with the simulated volume concentration falling however within the measurement uncertainties
envelope. As expected, for bins 4 and 5, the model is not capable of reproducing the observed PSD at distant areas since quite
significant underestimations have been already notified above sources (see Fig. 5a). The reduction of the settling velocity (i.e.,
URx runs, see Table 3) has negligible impact on the level of agreement between model and observations for bins 1 and 2,
moderate for bin3 while is determinant for the super-coarse (bin 4) and giant (bin 5) dust particles. Nevertheless, for achieving
the best model-observations matching, the necessary reduction (expressed in percentage) on the settling velocity is not constant
among the defined transport bins. Focusing on bins 4, the UR60 run (i.e., reduction of the settling velocity by 60%) outperforms
the other numerical experiments and focusing on bin 5 the UR80 run.
The overall comparison of the observed and modelled average PSDs is presented in Fig 8. We are considering all the
in situ airborne measurements and the WRF-L numerical outputs satisfying the defined spatiotemporal collocation criteria.
Error bars indicate the corresponding standard deviation. Figure 8a shows that the best model performance is found for the
UR80 experiments resembling satisfactory the bin 4 and bin 3/5 concentrations, respectively. These "artificial" reductions
translate to settling velocities equal ~0.066 for bin 3 (D=5.5-17 μm), ~0.32 m/s for bin 4 (D=17-40 μm) and ~1.88 m/s for bin
5 (D=40-100 μm). it is also reminded that for the same experiment it has been achieved the best agreement against the
FENNEC-PSD above dust sources (see Fig. 5 and the relevant discussion).
An alternative comparison between observations and model volume concentrations, for the selected AER-D samples
(each flight segment is denoted with different marker), has been performed and the obtained results, at each flight altitude, are
depicted in Figure 8b. More specifically, we calculate for each model experiment (denoted with different colour), the relative
differences (expressed in percentage) of the total dust volume concentration with respect to the in-situ measurements. In
addition, the corresponding differences (in percentage terms) that are representative for the altitudes spanning from near-
surface up to ~4.2 km are denoted with the vertical coloured dashed thick lines (WRF-L experiments). Those differences are
derived by averaging the relative differences of each flight segment. Overall, the model tends to underestimate the total dust
volume concentration (relative differences up to 100% in absolute terms) even though occasionally positive departures are
found, as indicated by the spread of the individual biases around zero. Nevertheless, the main finding from this analysis is that
the model-observation declations reduce when the settling velocity reduces too (i.e., URx runs). Among the WRF-L
experiments, the minimum biases (~5%) are obtained for the UR40 scenario (i.e. the vertical orange dashed line resides close
to zero). Through the inspection of the vertically resolved "behavior" of the individual runs, it is revealed that in some cases
the model-observation biases can be minimized for the UR60 and UR80 runs and this "variability" highlights the complexity
of the underlying mechanisms governing the suspension of airborne dust.

## 3.5 Dust vertical distribution

Figure 9(a) shows the profile of the mean extinction coefficient at 532 nm, provided by the LIVAS pure-dust product
(black line), and the profile of the mean extinction coefficients at 550 nm, provided by the CONTROL, UR20, UR40, UR60,
and UR80 experiments. The orange area indicates the standard deviation of the LIVAS profiles. Figure9(b) depicts the mean
absolute model bias with respect to LIVAS profiles for the different simulations and the vertical dashed lines show the
corresponding bias averaged over different altitudes. The mean LIVAS profile is provided by averaging the night-time profiles
over the region between 25.5oW to 12.5oE and 11.5oN to 35.5oN, during 5 to 25 August 2015. This area includes the main
dust sources that affected the vicinity of Cape Verde (Ryder et al., 2018) and the region of the dust outflow over the Ocean, as
well. The nightime profiles excel in accuracy over the daytime ones, due to the lower signal-to-noise ratio (SNR) during the
night. The model profiles are collocated in space and time with the LIVAS profiles, as described in Sect. 2.1.4 and the model
extinction coefficient is provided with the Eq.13.
The intercompared profiles are in good agreement, with the simulations falling well-within the variability of the dust
observations, although discrepancies are also present, especially close to the dust sources, in the nighttime boundary layer
(Fig.9(b) – region I), and within the upper free Troposphere (Fig. 9(b) – region III). The assessment of the different model
experiments against the ESA-LIVAS pure-dust product is performed in the region between 1.5 km and 6.4 km a.m.s.l. (Fig. 9
– region II), to avoid possible biases propagating into the analysis (i.e., complex topography and surface returns-region I, SNR
and tenuous aerosol layers – region II). According to the comparison of observations and simulations of the mean extinction
coefficient (Fig. 9(a)), the statistical overall analysis reveals that the UR40 experiment demonstrates a better performance
compared to LIVAS, reducing the mean bias close to zero. For the same experiment the minimum mean bias with respect to
the total volume concentration is achieved (see discussion of Fig.9b in Sect. 3.4). However, the UR80 experiment provides a
more constant (positive) bias with height, which suggests a better distribution of the dust mass in the vertical.

## 4 Discussion

The frequent presence of large desert dust particles (D>20 μm) far from their sources, is well established by numerous
observational studies over the last decade (van der Does et al., 2018; Liu et al., 2018; Ryder et al., 2013a, 2013b, 2018, 2019a;
Weinzierl et al., 2009, 2011, 2017). However, the processes that result in the particle retainment in the atmosphere, and
subsequently their travel at greater distances than predicted, remains unrevealed. In this study we extend the particle size range
applied in the transport parameterization of the GOCART-AFWA dust scheme of WRF, to include particles with diameters up
to 100 μm. The evaluation against airborne in situ observations of the size distribution shows that the concentrations of the
larger particles are underestimated, both above dust sources and distant areas. This suggests that there are atmospheric
processes that are not taken into account in the model simulations. We investigate the effect of reducing the settling velocity
of the dust particles due to these unknown processes, and we see that for a reduction of 60% and (especially for) 80%, the
simulations of the PSD in Cape Verde are improved with respect to the observations. The reduction of 80% corresponds to a
reduction in settling velocity of 0.0066 m/s for particles with D between 5.5 and 17 μm, which is double than the value reported
by Maring et al. (2003) for similar sizes. It should be noted though that Maring et al. (2003) derived this settling velocity using
observations that were taken with a five-year difference. Ginoux (2003), has also reported an improvement in model
simulations for a reduction in settling velocity of approximately 45% and 60%, for particles with diameters 10 to 30 μm.
Though, the differences in the model resolution, the dust scheme and the drag coefficient in Ginoux (2003) compared to this
study, could cause the different values of the required corrections in the settling velocities. The difference with the values
suggested herein, can mainly be attributed to the different drag coefficient used in Ginoux (2003), which results in lower
settling velocities for the spherical particles. Meng et al. (2022) performed a study, similar to this, where after reducing the
settling velocity by 13% for accounting for particles' asphericity based on Huang et al., (2020), performed sensitivity tests
reducing the dust particles' density from 2500 kg m$^{-3}$ to 1000, 500, 250 and 125 kg m$^{-3}$. They found that a decrease in the
modelled dust aerosol density by 10-20 times its physical value (2500 kg m$^{-3}$) is needed to improve the comparison between
the model and the long-range dust observations of coarse particles.  A 10 times reduction in particles' density is almost equal
to a 90% reduction in the settling velocity (starting from the Clift and Gauvin (1971) drag coefficients and assuming conditions
of U.S. Standard Atmosphere, Fig. S1). It is clear that a huge reduction in the settling velocity in both the Meng et al., (2022)
methodology and this work is required, although the physical processes occurring to explain this reduction are not clear.

One of the processes proposed in the literature to explain the longer atmospheric lifetimes of large mineral dust particles is the particle asphericity. Ginoux (2003) compared randomly-oriented prolate spheroids and spheres of the same cross section. They showed that spheroids fall slightly slower than their spherical counterparts, with their difference being negligible for spheroids with aspect ratio values less than 5. Huang et al. (2020) compared randomly-oriented ellipsoids and spheres of the same volume. They showed that ellipsoids fall around 20% slower than spheres. Mallios et al. (2020) compared prolate spheroids and spheres of the same maximum dimension, and of the same volume. Moreover, they did not assume randomly-oriented particles, but particles of specific orientation (horizontal and vertical). They showed that the results of the comparison change when the maximum dimension or the volume-equivalent size is used in the comparison. Prolate spheroids, with aspect ratio values in the range of 1.4-2.4, fall slower than spheres of the same maximum dimension, regardless of orientation, with the relative difference between the settling velocities reaching the value of 52%. On the other hand, prolate spheroids, in the same aspect ratio value range, fall faster than spheres of the same volume, regardless of orientation. The comparison with in situ observations of the maximum dimension of particles is not so common, since most of the in-situ measurements do not provide the sizing of the particles in terms of their maximum dimension, with some exceptions, as e.g. the observations shown in van der Does et al. (2016) of individual giant mineral particles (larger than 100 μm in maximum dimension).

All the above show that more work is needed for the definite and accurate quantification of the particle asphericity effect on their settling. Nevertheless, there are indications pointing that aspherical particles remain in the atmosphere longer, and that asphericity can be one of the reasons for the differences between the modelling results and the observations.

Another process that can influence mineral dust settling has to do with the electrical properties of dust particles. The dust particles are charged in the atmosphere either due to the attachment of atmospheric ions on them (Mallios et al. 2021b) or/and due to collisions, a process known as triboelectric effect (Ette, 1971, Eden and Vonnegut, 1973, Mills, 1977, Jayaratne, 1991, Mallios et al., 2022). Moreover, there is a large-scale atmospheric electric field, due to the potential difference between the lower part of the Ionosphere and the Earth's surface (Rycroft et al., 2008). The electric field is modified by ion attachment process (Mallios et al. 2021b) or by the charge separation caused by updrafts (Krauss et al., 2003). Therefore, electrical forces are generated that might influence the particle settling process by balancing the gravity or changing the particle orientation. The quantification of the particles' electrical properties is still an open question

Another possible source of error in the gravitational losses simulated by the model as proposed by (Ginoux, 2003) is the numerical diffusion in the advection equation of gravitational settling. Since in the GOCART-AFWA dust scheme of WRF (and WRF-L) a first-order upwind scheme is adapted for the gravitational losses, which is rather diffusive (Versteeg and Malalasekera, 2007), an investigation of the possible improvement on the results by the replacement of the scheme with a less diffusive would be of interest. A possible limitation of this study is the accuracy of the PSD which is used for the distribution on the model transport bins of the emitted fluxes. The simplification in the assumption that the shape of the PSD at 1km above the sources remains unchanged in lower heights near the ground, could possibly introduce errors in the representation of the presence of dust particles aloft.

In any case, the proposed scheme presented here, provides a useful tool for the investigation of the physical processes
in the transport of coarse and giant particles, along with their impacts on other physical processes in the atmosphere, such as
ice nucleation and radiation interactions. The artificial reduction in the settling velocity is not attributed to a known physical
mechanism (although results from the past literature reveal some candidates that can give results on the same order of
magnitude). Thus, despite the encouraging results, more research is needed towards understanding the physical or numerical
processes driving this finding, including the estimation of the impact of non-spherical particles, electricity, the radiation impact
on thermodynamics and the disturbance of the mass balance due to the numerical diffusion.
**5 Summary and conclusions**
In the current state-of-the-art atmospheric dust models, several physical processes governing dust life cycle
components are not well represented or they are not included in the relevant parameterization schemes. This drawback, along
with the lack of knowledge on the underlying mechanisms, results in the failure of the numerical simulations to reproduce
adequately the long-range transport of super-coarse and giant mineral particles, as it has been justified via their evaluation
versus sophisticated dust observations. Among the model limitations, well documented in literature, one of the critical is the
neglect of mineral particles with diameters larger than 20 μm, under the erroneous assumption that they deposit quickly after
their emission.
In the current study, we modify the transport particle size distribution in WRF, expanding at size ranges up to 100
μm in diameter, by constraining the shape of the modelled PSD with the observed one above dust sources, acquired in the
framework of the FENNEC 2011 campaign. A novelty of our work constitutes the upgrade of the drag coefficient, determining
the settling velocity of dust particles, for accounting realistic dust particles sizes (Re < $10^5$), opposite to what is assumed in the
traditional Stokes' theory. After optimally tuning the CONTROL run, we performed a series of sensitivity experiments in
which the settling velocity has been reduced, aiming to artificially resemble the real forces acting on particles moving vertically
and counteract gravitational settling. Our period of interest spans from 5[th] to 25[th] August 2015, when the AER-D campaign
took place in the surrounding area of Cape Verde, residing in the core of the "corridor" of the Saharan dust transport along the
Tropical Atlantic Ocean. In our experiments, the simulation domain covers most of the Sahara Desert (encompassing the most
active dust sources worldwide) and the eastern sector of the Tropical Atlantic Ocean (receiving large amounts of mineral
particles from the nearby Saharan dust sources). The dust-related numerical outputs produced by the CONTROL and URx
(referring to the reduction of the settling velocity by 20%, 40%, 60% and 80% and it is expressed by the term x) experiments
are evaluated against the LIVAS satellite datasets providing pure dust extinction vertical profiles. Nevertheless, special
attention is given on the evaluation of the WRF-L PSD against airborne in-situ measurements acquired in the framework of
the AER-D campaign.
Based on our results, in the CONTROL experiment, the model tends to underestimate the dust volume concentration
of coarse and giant dust particles (FENNEC) since the very early stage of dust transport, when the emitted mineral particles

are uplifted at 1 km above the sources. Subsequently, the initially obtained model underestimation becomes more pronounced, against those measured in AER-D, particularly for the super-coarse (bin 4, diameters from 17 to 40 μm) and giant (bin 5, diameters from 40 to 100 μm) dust particles, in the vicinity of Cape Verde (i.e., downwind region). Our findings are in line with the already stated underestimation of the presence of coarse and giant dust particles' presence during their long range dust transport. Nevertheless, when we gradually reduce the settling velocity (URx runs) the model performance steadily improves. Overall, among the numerical experiments, the best match of the simulated and the observed PSDs is achieved for the UR80 scenario (i.e., reduction of the settling velocity by 80%), thus highlighting the misrepresentation or the absence of forces, within the model parameterization schemes, acting on dust particles and counteract gravitational settling. Through the case-by-case inspection, it is revealed that the UR60 and UR40 scenarios can also occasionally provide the optimum model-observations agreement thus highlighting the complexity of the real physical processes that regulate dust particles' settling velocity and suspension. From the evaluation of the vertically resolved simulated dust extinction against the corresponding measurements from the LIVAS dataset, it is revealed that for the UR40 run the model-observations are minimized (oscillating around zero) whereas the UR80 run outperforms in reproducing the vertical structure of the dust layers within the Saharan Air Layer. Summarizing, our work demonstrated an innovative approach in order to overcome existing drawbacks of the atmospheric-dust models towards improving the simulations of dust transport along the Tropical Atlantic Ocean. There are several candidate mechanisms, along with inappropriate definition and treatment of mineral particles in the parameterization schemes, hampering models in reproducing adequately the observed dust patterns. Despite our encouraging results, there are many mandatory steps towards upgrading the current state-of-the-art atmospheric dust models in anticipation of an optimum assessment of the multifaceted role of dust aerosols within the Earth-Atmosphere system.

**Author Contributions**: ED, VA, and AT design the study; SM guided ED on the methodology for the replacement of the drag coefficient. AT provided useful assistance on the treatment of airborne observations. CR provided the data from the airborne in situ measurements and provided useful information about the instrumentation methods. ED developed the code, performed the simulations and analyzed the results. AG and CR consulted ED on the methodology of in situ and WRF datasets. VA, EM and EP provided the LIVAS dataset, lead the collocation methodology and helped on the interpretation of the results. ED plotted the model and observation data (apart from LIVAS). EP treated and plotted LIVAS data; ED wrote the manuscript draft; VA, AT, AG, EP, SM, CS, SS, EM, CR, DB and PK provided critical feedback and reviewed and edited the manuscript.

**Funding:** This research was supported by the project "PANhellenic infrastructure for Atmospheric Composition and climatE change" (no. MIS 5021516), which is implemented under the action "Reinforcement of the Research and Innovation Infrastructure", funded by the "Competitiveness, Entrepreneurship and Innovation" Operational Programme (NSRF 2014–2020) and co-financed by Greece and the European Union (European Regional Development Fund). Support was also provided by D-TECT (Grant Agreement 725698) funded by the European Research Council (ERC). ED would like to acknowledge funding by Greece and the Stavros Niarhos Foundation (SNF). CLR was funded by NERC grant reference NE/M018288/1.

EM was funded by the European Research Council
(grant no. 725698, D-TECT) and by a DLR VO-R young investigator group and the Deutscher Akademischer
Austauschdienst (grant no. 57370121). AG acknowledges support by the Hellenic Foundation for Research and Innovation (H.
F. R. I.) under the "2nd Call for H. F. R. I. Research Projects to support Post-Doctoral Researchers" (project acronym:
ATLANTAS, project number: 544).

**Acknowledgements:** This work was supported by computational time granted from the National Infrastructures for Research
and Technology S.A. (GRNET S.A.) in the National HPC facil–ty - A–IS - under project ID pa210502-TRAP-P. We thank the
PANhellenic GEophysical observatory of Antikythera (PANGEA) for providing access to the LIVAS data used in this study
and their computational center. The National Centers for Environmental Prediction (NCEP) is gratefully acknowledged for the
provision of the Global Forecasting System (GFS) operational analyses and the real time global (RTG) sea surface temperature
(SST) analyses. We would like to thank the NASA CALIPSO team and NASA/LaRC/ASDC for making the CALIPSO
products available, which have been used to build the LIVAS products, and ESA, who funded the LIVAS project (contract no.
4000104106/11/NL/FF/fk).

**Data Availability:** The model outputs and the data used for the analysis are available upon request from Vassilis Amiridis
(vamoir@noa.gr) and/or Eleni Drakaki (eldrakaki@noa.gr). The LIVAS dust products are available upon request from Vassilis
Amiridis (vamoir@noa.gr), Emmanouil Proestakis (proestakis@noa.gr), and/or Eleni Marinou (elmarinou@noa.gr).

**Code Availability:** The source code of WRF-L is available upon request from Vassilis Amiridis (vamoir@noa.gr) and/or Eleni
Drakaki (eldrakaki@noa.gr).

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

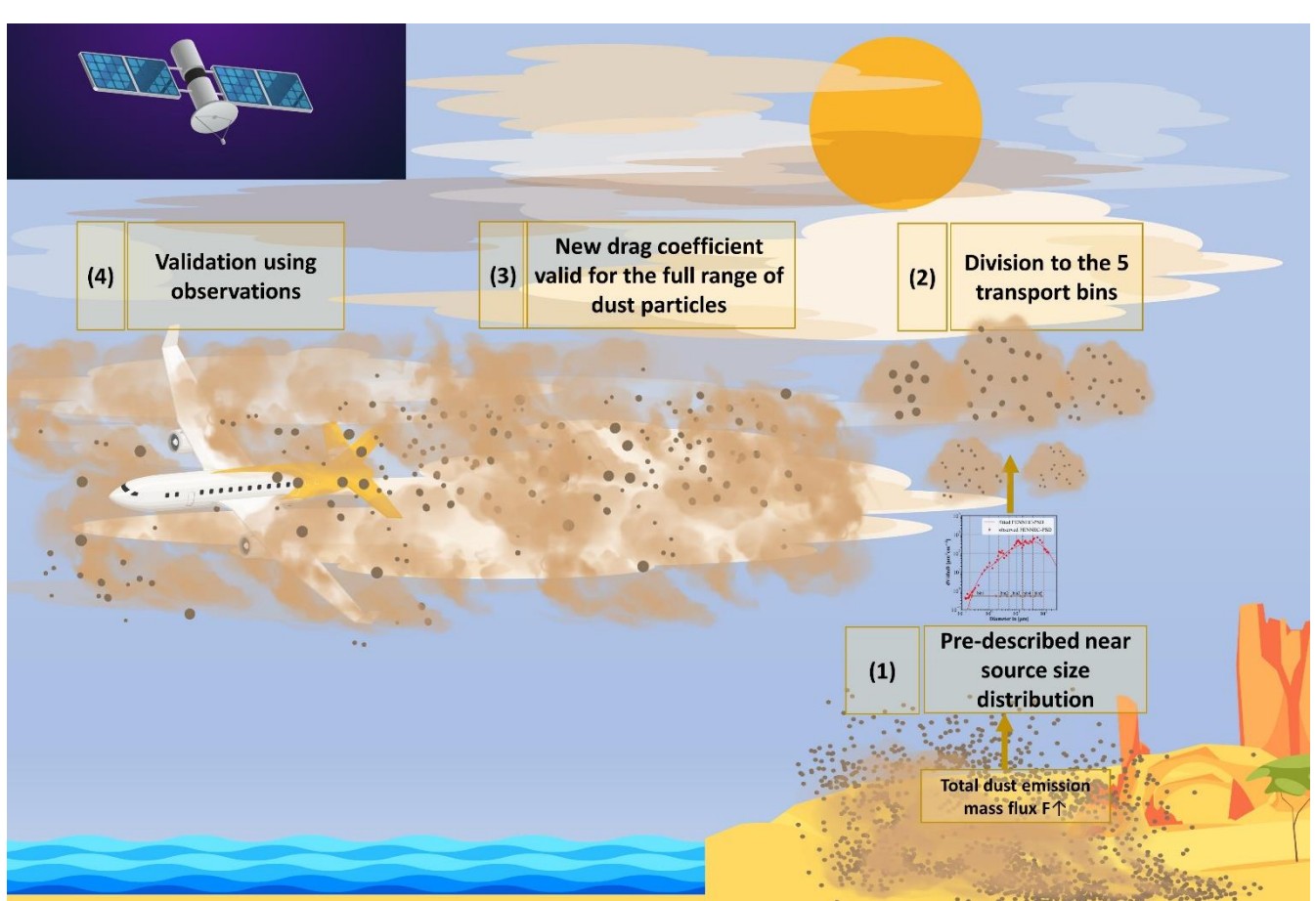


**Figure 1: The structure of the presented work. Steps (1), (2) and (3) correspond to the appropriate modifications implemented in the WRF-Chem GOCART-AFWA dust scheme, for the inclusion of the giant dust particles and the development of WRF-L. Step (4) refers to model validation activities.**

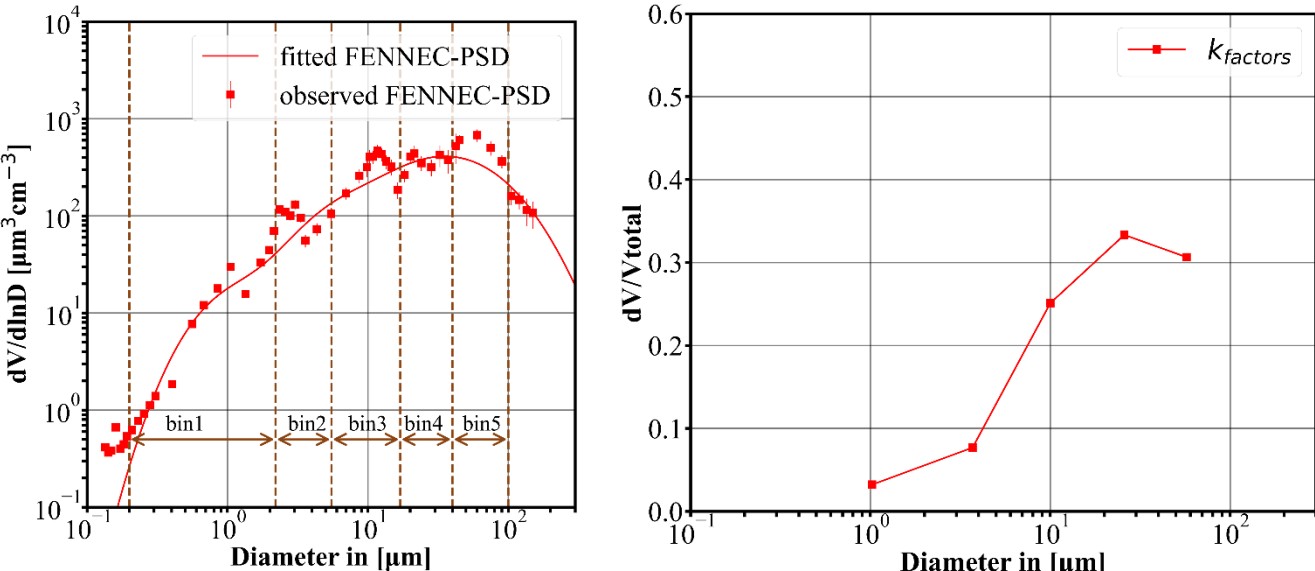


**Figure 2: Prescribed dust size distribution used in the WRF-L for the distribution of total dust mass to the transport model size bins: (a) "observed FENNEC-PSD" (μm3cm-3) (red squares), and the respective "fitted FENNEC-PSD" (red solid line). The "observed FENNEC-PSD" corresponds to the PSD observations at 1km, obtained by averaging profile measured data of freshly uplifted dust cases, over 500m. The arrows indicate the model transport size bins in WRF-L. Error bars indicate the standard deviation of the observed values (b) The $k_{factors}$ of the transport size bins calculated based on "fitted FENNEC-PSD", provide the mass fraction of the emitted dust for each bin.**

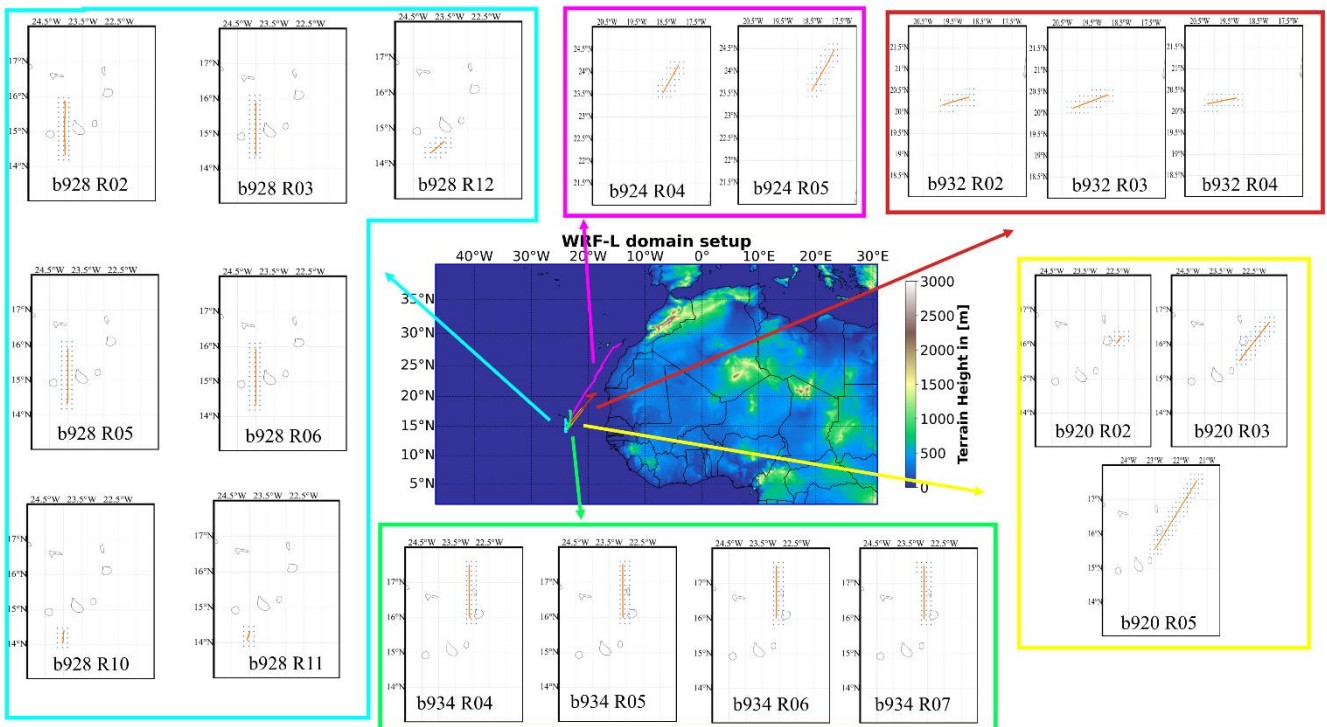

Figure 3: Domain and topography map of the WRF-L model simulations, with a horizontal grid spacing of 15km, and 70 vertical levels. The tracks of the AER-D flights, used in this study (b920, b924, b928, b932 and b934), are depicted in the central plot with different colors. In the surrounding maps, the orange dots indicate the aircraft tracks of each flight RUN. The blue dots correspond to the collocated model grid points.

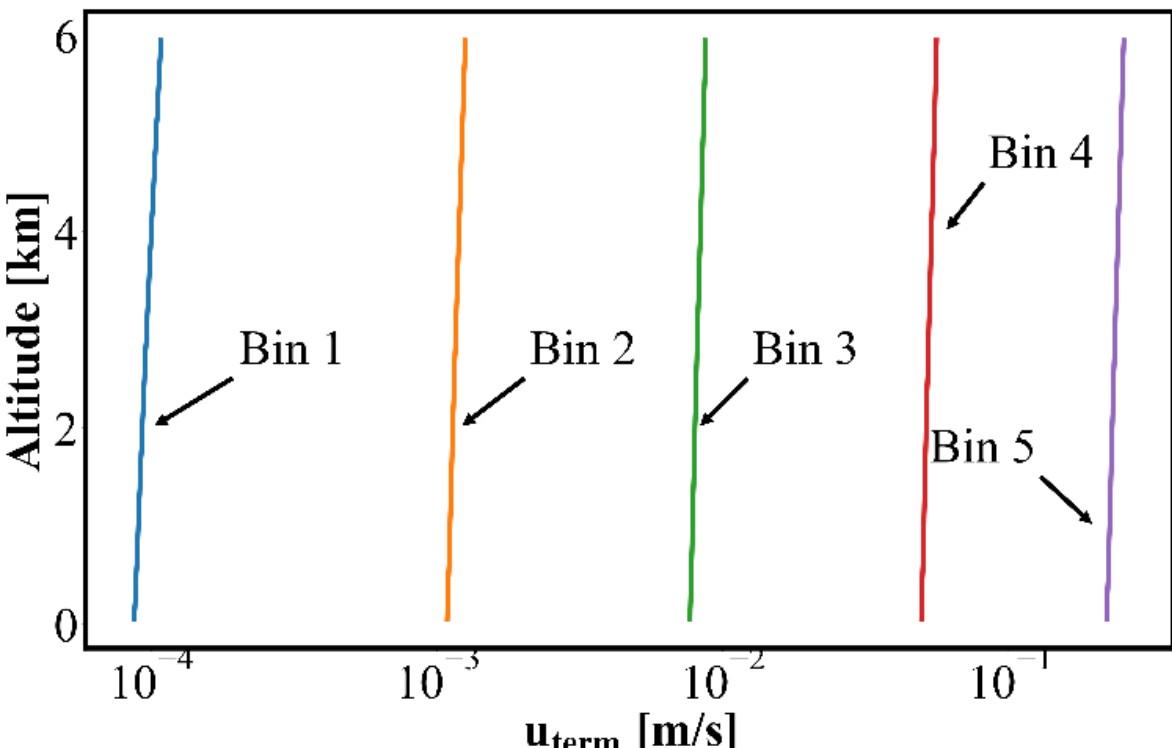


**Figure 4: Terminal velocities of the CONTROL experiment, averaged for the simulation time and the domain. Each**
**colored line corresponds to one of the new model size bins, with blue: Bin 1, orange: Bin 2, green: Bin 3, red: Bin 4 and**
**purple: Bin 5.**

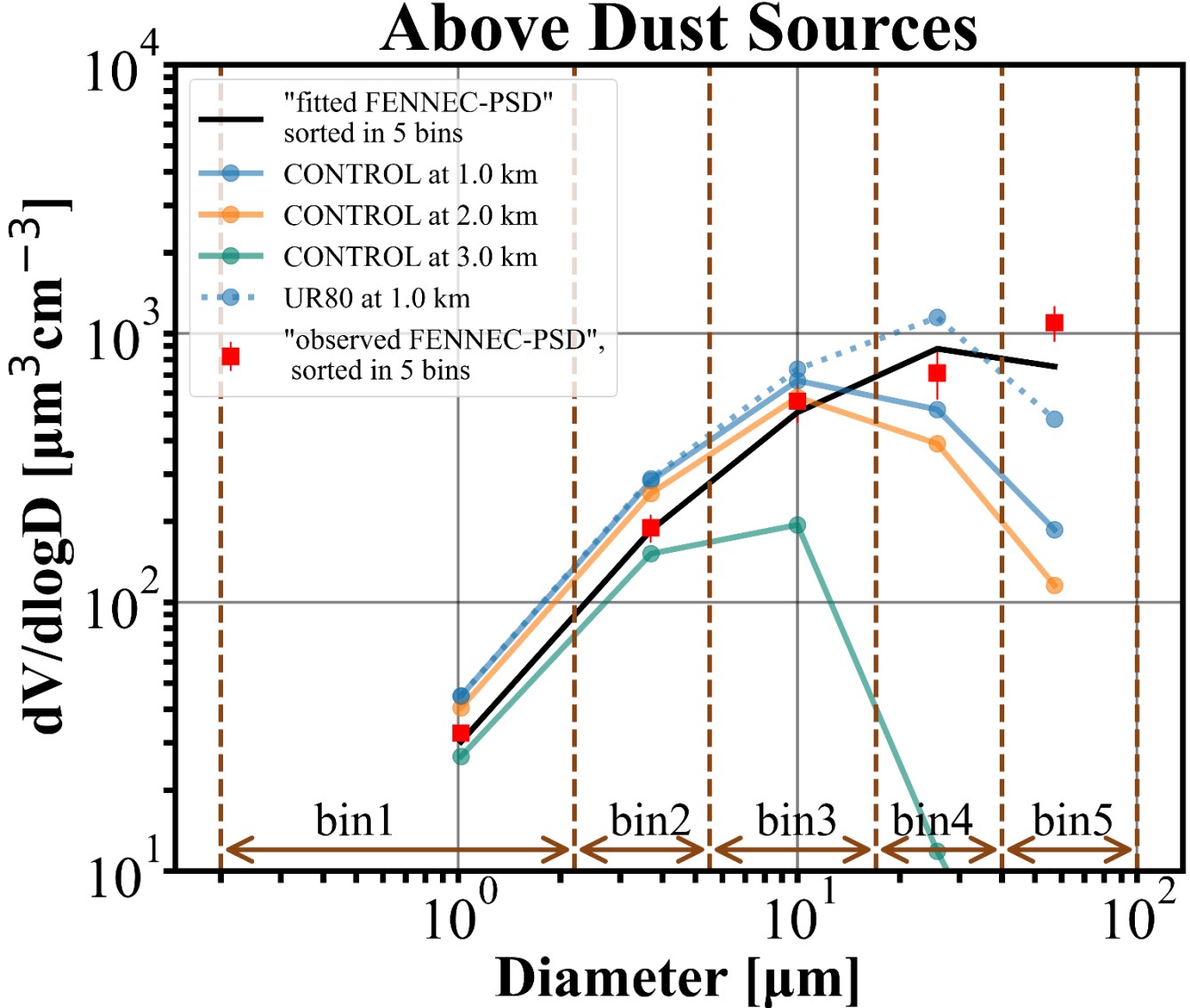

Figure 5: Dust size distribution above an emission model grid point (latitude=24.9o and longitude=9.2o) in Mali, on 11/08/2015 at 14UTC. Blue solid line: the dust PSD of the CONTROL run interpolated at 1 km altitude above the dust source, orange solid line: the dust PSD of the CONTROL run interpolated at 2 km altitude above dust source, green solid line: the dust PSD of the CONTROL run interpolated at 3 km altitude above dust source, blue dotted line: the dust PSD of the UR80 run interpolated at 1 km altitude above the dust source and red squares: the "observed FENNEC-PSD" at 1 km altitude (sorted in 5 bins), black squares the "fitted FENNEC-PSD" at 1km (sorted in 5 bins) which has been used for the distribution of the model emission to the five size bins.

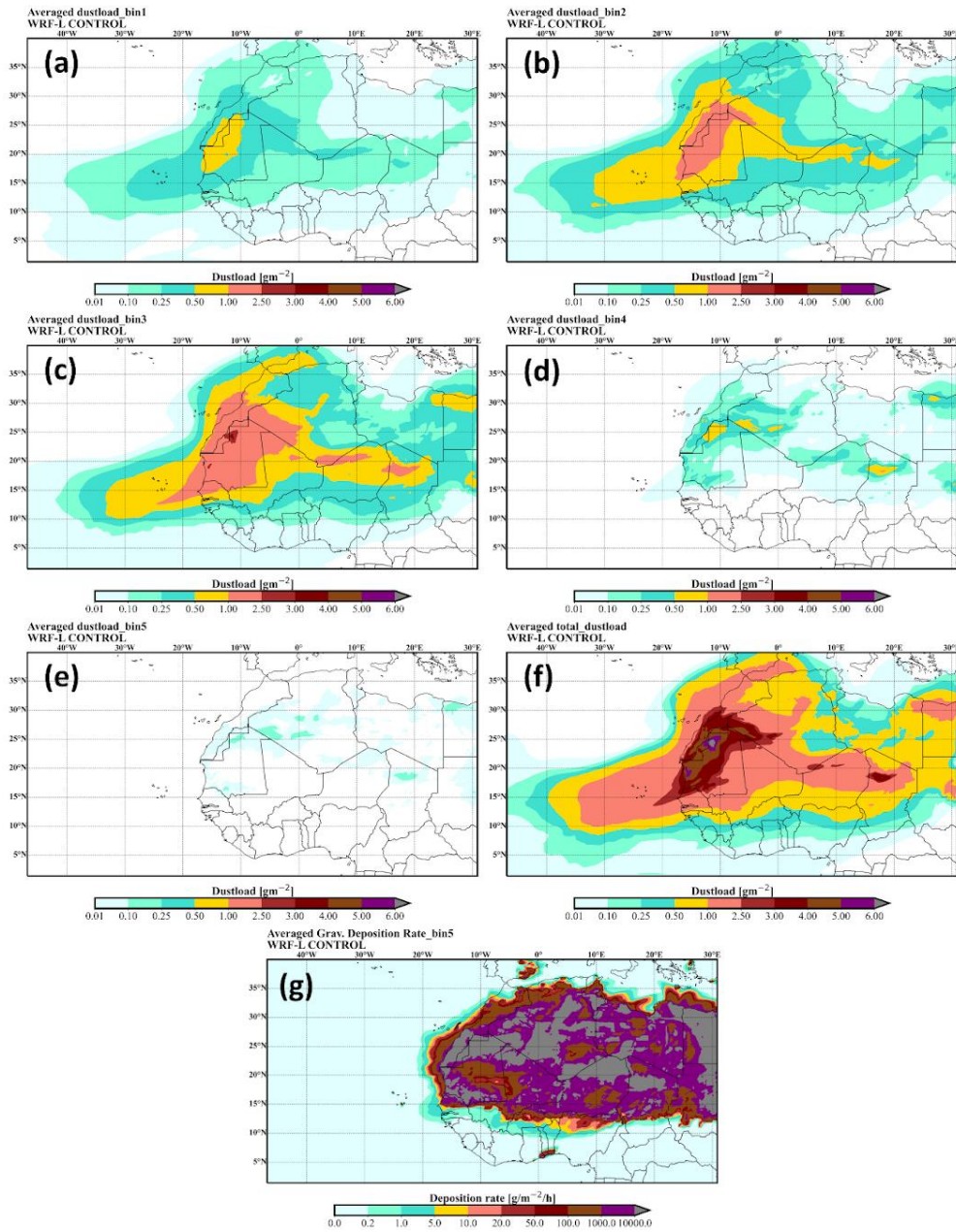

923

**Figure 6: The dust load provided by the model, averaged for the whole simulation period, for (a) bin 1, (b) bin 2, (c)**

**bin 3, (d) bin 4, (e) bin 5, and (f) the whole range of the PSD. The dust load is in g/m². (g) The gravitational deposition**

**rate for bin 5 in g/m²/h.**

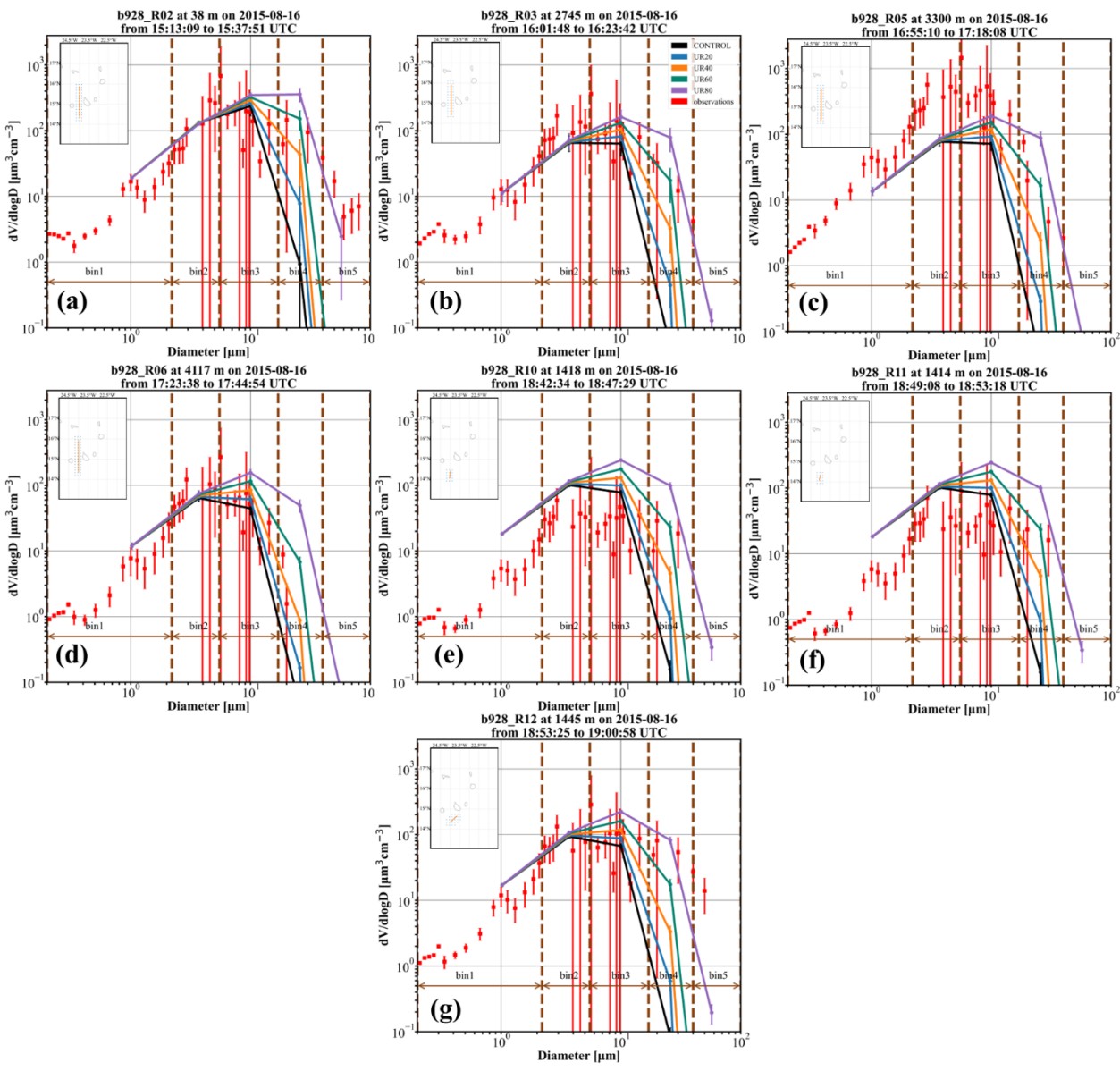

927

**Figure 7: Modeled and observed dust PSD of flight b928, for straight-level-runs (a) R02, (b) R03, (c) R05, (d) R06, (e) R10, (f) R11 and (g) R12. The in situ observations are shown with red squares (along with the total instrumentation error). The collocated modeled PSDs are shown with lines, for the CONTROL run (black), UR20 (blue), UR40 (orange), UR60 (green), and UR80 (purple) and the corresponding standard deviation with the associated error bars. The brown vertical lines indicate the limits of the model size bins. The inlet maps show the flight segment track and the collocated model grid points.**

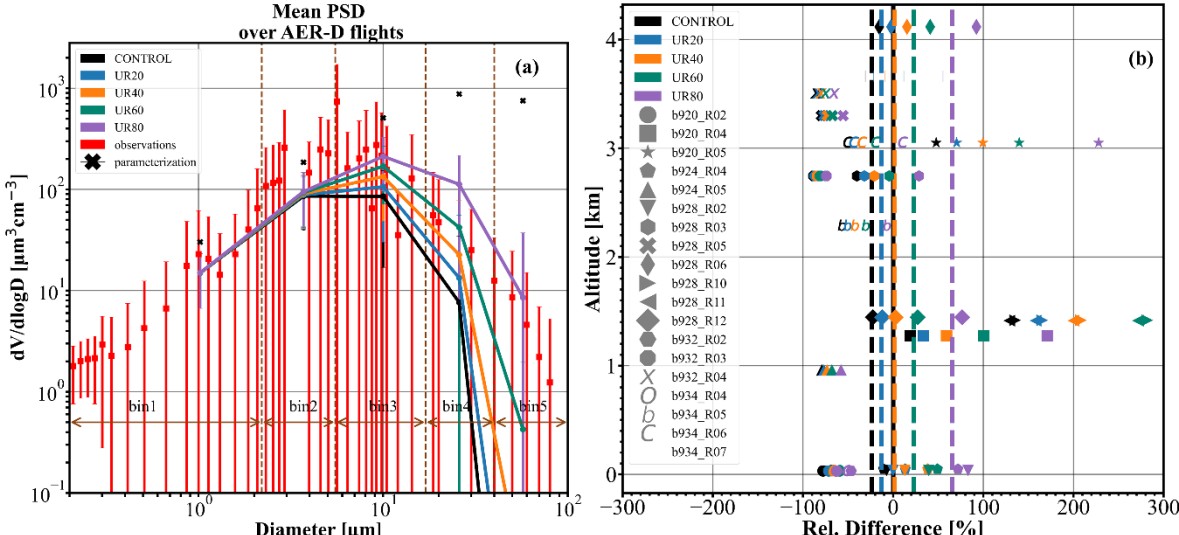

**Figure 8: (a) Mean PSD of AER-D/ICE-D campaign. The observations are shown with red squares, whereas the simulations are shown with solid lines for the CONTROL run (black), UR20 (blue), UR40 (orange), UR60 (green), and UR80 (purple). (b) The relative difference between the observations and the model simulations of the total volume of dust particles, at different altitudes. The observations from different flight segments (i.e., b920 R02, b920 R04, b920 R05, b924 R04, b924 R05, b928 R03, b928 R05, b928 R06, b932 R02, b932 R03, b932 R04, b934 R04, b934 R05, b934 R06, and b934 R07) are denoted with different markers. The average relative difference of the observations and the simulations are denoted with dashed lines, for the CONTROL run (black), UR20 (blue), UR40 (orange), UR60 (green), and UR80 (purple).**

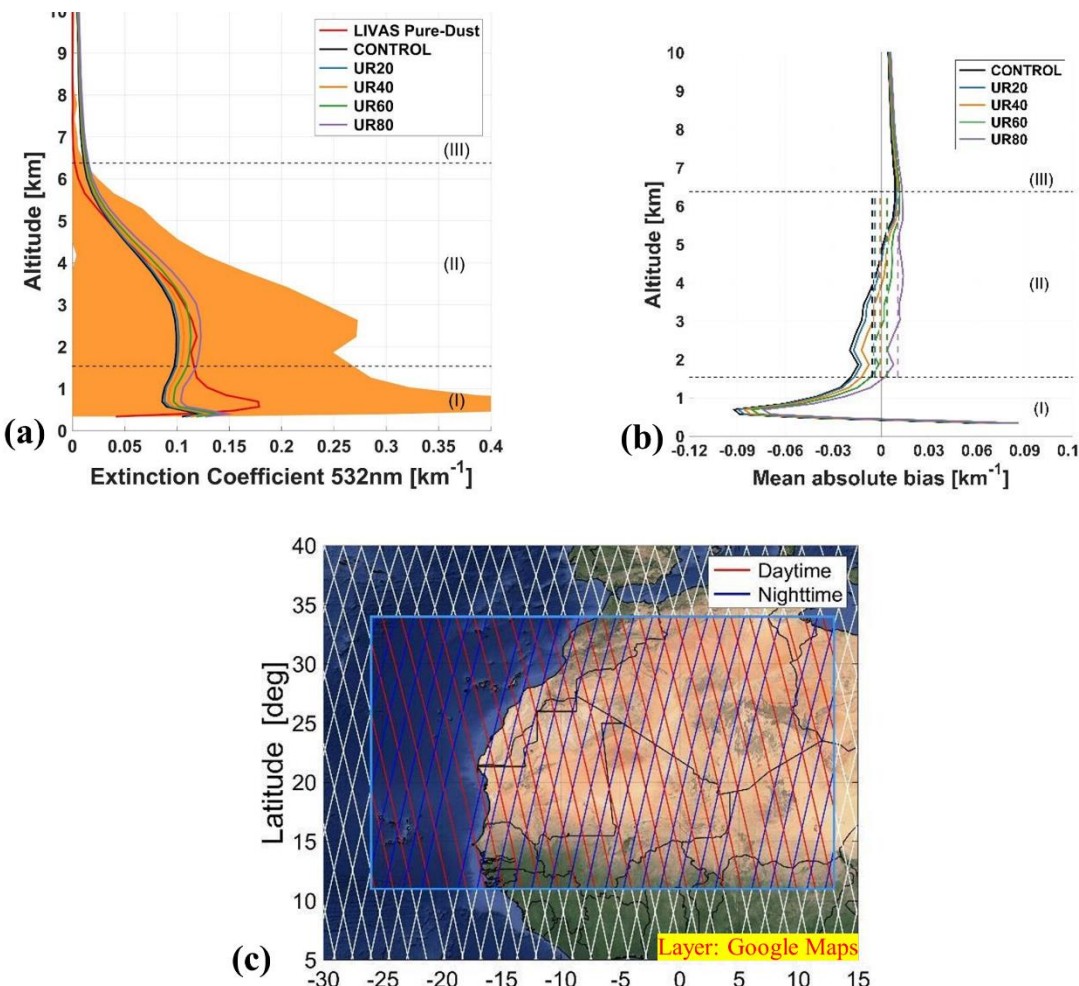

946

Figure 9: (a) Profile of the mean extinction coefficient at 532 nm, by LIVAS pure-dust product (black red line), and
profiles of the mean extinction coefficient at 532 nm simulated from the different experiments of Table 3 (CONTROL,
UR20/40/60/80). The orange shading indicates the standard deviation of the LIVAS profile averaging. (b) The mean
absolute biases between the LIVAS profile and the simulated profiles from the different experiments, in the domain of
interest, between 05/08/2015 and 25/08/2015. The vertical dashed lines are the mean absolute bias between the LIVAS
profile and the simulated profiles from the different experiments averaged over the altitudes of region II. (c) The
domain of interest and the daytime (red) and nighttime (blue) CALIPSO overpasses. The vertical dashed lines are the
mean absolute bias between the LIVAS profile and the simulated profiles from the different experiments averaged over
the altitudes of region II. Layer: google maps background.



**Table 1 Size ranges and properties of model size bins in the default WRF-GOCART-AFWA scheme**

| WRF-GOCART-AFWA | | | | | |
|---|---|---|---|---|---|
| **Bins** | **1** | **2** | **3** | **4** | **5** |
| $D_{lo} - D_u$ **(μm)** | 0.2-2.0 | 2.0-3.6 | 3.6-6.0 | 6.0-12.0 | 12.0-20.0 |
| $D_{eff}$ **(μm)** | 1.46 | 2.8 | 4.8 | 9.0 | 16.0 |
| $\rho_p$ **(g cm⁻³)** | 2.5 | 2.65 | 2.65 | 2.65 | 2.65 |
| **WRF-L** | | | | | |
| **Bins** | **1** | **2** | **3** | **4** | **5** |
| $D_{lo} - D_u$ **(μm)** | 0.2-2.2 | 2.2-5.5 | 5.5-17.0 | 17.0-40.0 | 40.0-100.0 |
| $D_{eff}$ **(μm)** | 1.02 | 3.7 | 10.0 | 25.8 | 57.2 |
| $\rho_p$ **(g cm⁻³)** | 2.5 | 2.65 | 2.65 | 2.65 | 2.65 |


**Table 2 Configuration parameters of the WRF-L runs**

| Parameterization | Scheme | Parameterization | Scheme |
|---|---|---|---|
| **Surface Model** | Noah (Chen and Dudhia, 2001) | sf_surface_physics | 2 |
| **Surface Layer** | Monin-Obukov-Janjic (Janić, 2001) | sf_sfclay_physics | 2 |
| **Radiation (SW and LW)** | RRTMG (Iacono et al., 2008) | ra_sw(lw)_physics | 4 |
| **Microphysics** | Morrison 2-moment (Morrison et al., 2005) | mp_physics | 10 |
| **Cumulus** | Grell-3 (Grell and Dévényi, 2002) | cu_physics | 5 |
| **Boundary Layer** | MYNN 2.5 (Nakanishi and Niino, 2006) | bl_pbl_physics | 5 |
| **Chemistry** | GOCART simple (Ginoux et al., 2001; LeGrand et al., 2019) | chem_opt | 300 |
| **Dust Scheme** | AFWA (LeGrand et al., 2019) | dust_opt | 3 |

**Table 3 Experimental runs that performed in this study**

| Experiment | Code |
|---|---|
| **CONTROL** | WRF-L |
| **UR20** | WRF-L with reduced settling velocities by 20% of their settling velocity |
| **UR40** | WRF-L with reduced settling velocities by 40% of their settling velocity |
| **UR60** | WRF-L with reduced settling velocities by 60% of their settling velocity |
| **UR80** | WRF-L with reduced settling velocities by 80% of their settling velocity |

**Table 4: Lognormal ($\frac{dV}{dlnD} = \frac{V_{tot}}{\sqrt{2\pi}ln\sigma_g} exp\left(-\frac{(lnD_v - lnD)^2}{2(ln\sigma_g)^2}\right)$) mode parameters of the fitted FENNEC-PSD. Diameters are**
**given in $[\mu m]$ and volume concentrations in $[\frac{\mu m^3}{cm^3}]$:**

| Modes | 1 | 2 | 3 | 4 | 5 |
|---|---|---|---|---|---|
| $V_{tot}$ | 15.16 | 27.07 | 169.32 | 310.5 | 563.3 |
| $D_v$ | 1.0 | 2.5 | 7.0 | 22.0 | 50.0 |
| $s_g$ | 1.8 | 2.0 | 1.9 | 2.0 | 2.15 |
