# Peer review of "Modelling coarse and giant desert dust particles"

_Atmospheric Chemistry and Physics, 2022_

## Author Comment (AC1)

This study investigates the incorporation of coarse and giant desert dust particles (with diameter greater than 20  $\mu$ m) in the WRF model, together with the GOCART aerosol model and the AFWA dust emission scheme. The authors implemented a number of extensions to the original model. More specifically, they used a prescribed dust particle size distribution for emitted dust particles at the source based on in situ measurements from the FENNEC campaign and employed 5 size bins with diameters up to 100  $\mu$ m (corresponding to giant particles). Moreover, they implemented an updated drag coefficient that applies to the above bins and is representative of high values of Re number. The simulations were performed from 29 July to 25 August 2015. The model output were validated against various observational datasets.

The article is well written and promotes the research in the modelling of the desert dust. The use of English is excellent and the conclusions are supported by the results. It is suggested to accept this article for publication after some minor corrections are performed.

The recognition of our work from the reviewer is much appreciated. We would like to thank him/her for taking the necessary time and effort to review our manuscript. We sincerely appreciate all your valuable comments and suggestions. The manuscript has been revised considering all the suggestions raised by the reviewer.

**Suggested corrections:**

Section 2.1.3: please include a) whether the vertical levels (line 220) were defined by WRF or by the authors (providing how you chose them in the latter case), b) which UTC time the original initialization/each was chosen for reinitialization (line 221), c) some more detailed information about the model results that you used from each 84 hour run (i.e. whether you removed the first 12 hours of each run due to model spin-up and utilized the rest; line 221), d) the topography and land-use datasets, e) whether the seasurface temperatures were updated from GFS-FNL analyses every 72 hours at the initial time of each run or every 6 hours together with the lateral boundary conditions.

We agree with this comment and we have incorporated the reviewer's suggestion throughout Section 2.1.3, explaining that the specific heights of the vertical levels are defined by the model. The sea surface temperatures in the model acquired by the NCEP daily SST analysis (RTG\_SST\_HR) are updated every six hours along with the lateral boundary conditions. Each 84-hour run was initialized at 12 UTC and the first 12 hours were removed accounting for the model spin-up. Topography is interpolated from the 30s Global Multi-resolution Terrain Elevation Data 2010 (GMTED2010, Danielson and Gesch, (2011)). We use land-data based on Moderate-resolution Imaging Spectroradiometer (MODIS) observational data modified by the University of Boston (Gilliam and Pleim, 2010). Hence based on our reply we modified the Sect 2.1.3 of the original manuscript (line 216, page 8):

"Using the WRF-L code, we first run the CONTROL experiment. Our simulation period coincides with the AER-D experimental campaign (29/7 - 25/8/2015) for a domain bounded between the 1.42°N and 39.99°N parallels and stretching between the 30.87°W and 46.87°E meridians (Fig. 3). The simulation area encompasses the major Saharan sources also including the downwind areas in the eastern Tropical Atlantic. We use an equal-distance grid with a spatial grid spacing of 15 km x 15 km consisting of 550 × 300 points whereas in vertical, 70 vertical sigma pressure levels up to 50 hPa are utilized. The simulation period consists of nine 84-hour forecast runs, which are initialized at 12 UTC, using the 6-hour Global Forecast System Final Analysis (GFS - FNL) reanalysis product, available at a 0.25°x0.25° spatial resolution. The sea surface temperatures, acquired by the NCEP daily global SST analysis (RTG\_SST\_HR), are updated every six hours along with the lateral boundary conditions. From each 84-hour cycle, the first 12 hours are discarded due to model spin up. Likewise, the first week of the simulation served as a spin-up run for the accumulation of the background dust loading and it is excluded from the analysis."

**Line 369-373: Have you validated the simulated upper air wind field, e.g. using ERA5? Western Africa is characterized by a complex wind regime. There is a large area with pink colors (i.e. dust) in area B of Figure 7f. Therefore, the dust errors may be also due to erroneous wind field.**

The reviewer raises an important issue regarding how a possible wind speed bias can affect the emission. Menut, (2008) quantified the impact of the meteorological data forcing (using either NCEP or ECMWF as initial/boundary conditions) above Sahara sources and reported that the difference between the two emission fluxes can reach a factor of 3. Moreover, they noted that this difference is not systematic and no conclusion was made of which dataset overperforms. Following the reviewer's suggestion, we performed a validation analysis of the WRF-L upper wind fields (i.e., at 300 and 500 hPa) versus ERA5, both reprojected at a common grid (0.25 x 0.25 spatial resolution). The obtained results for the two pressure levels, on 5th August 2015 at 00 UTC, are illustrated in the Figure below. It is evident that the two models produce similar meteorological patterns with deviations only on the wind speeds. Focusing on the latitudinal band (10-25°N) where the Saharan dust is transported over the Tropical Atlantic Ocean, mainly positive WRF-ERA5 declinations are recorded over the W. Sahara while the opposite is revealed over the outflow regions. This differences in the two models above land are almost consistent throughout the whole simulation. In terms of magnitude lie mostly in the range of 2-8 m/s (in absolute terms) at 500 hPa and they are slightly higher at 300 hPa.

Figure R1: WRF-L wind fields at (a) 500 and (b) 300 hPa, and wind speed differences with respect to ERA5 wind fields at (c) 500hPa and (d) at 300 hPa.

Deviations in the wind fields can impact both the emission and the transport of dust. The link between winds and produced emissions and transport is rather a complex issue and needs a more thorough investigation, which is beyond the scope of this article. More specifically, regarding the accuracy of the atmospheric models' forecasts, among the possible reasons could be the induced uncertainties in the wind fields of the global datasets, which are used as initial and boundary conditions. In the global datasets, the assimilation of observations and measurements assists models in reducing their errors. Please note that according to RC3 reviewer comments, to avoid any confusion in the reader, the part related to Fig. 7 of the original manuscript has been removed.

**Technical corrections:**

**Line 23: "… diameters of 5.5-17 μm …"**

---

## Author Comment (AC2)

**This paper addresses the by now seemingly well established underestimation of coarse and giant dust particles by large-scale models. This is an important topic, as these particles are much more abundant than previously thought, such that models could be missing important effects on radiation, clouds, and biogeochemistry. The present paper tries to address this issue by using in situ measurements of dust size distribution over the North African source regions to parameterize the sizes of emitted dust in the WRF-Chem model and then comparing the results against. They find that the deposition velocity of**
**particles must be greatly reduced in order for the model to match measurements further from source regions, which further confirms previous findings in the literature that coarse dust deposits too quickly in models.**

**Overall, this is a useful contribution to the literature. I did find a series of issues with the description of the methods and results. None of them are serious enough to preclude publication and I'm hopeful that a next version would be suitable for publication. Nonetheless, major revisions are required.**

We appreciate the positive feedback of the reviewer and we would like to thank him/her for the effort and expertise that he/she contributed towards reviewing our manuscript. We are grateful for the insightful comments and we tried to incorporate changes to reflect the provided suggestions.

**Specific comments:**

- **I think the paper should be clearer about the actual objective of the paper is or the scientific question it addresses. If this is just to "extend the parameterization the mineral dust cycle in the GOCART-AFWA dust scheme of WRF4.2.1 to include also coarse and giant particles" then this is pretty narrow and perhaps better suited for GMD or a similar journal. But it seems that the authors also investigate the reasons for why coarse and giant dust is underestimated by models, finding that particles settle much too fast in the model. I would suggest making this objective of the paper clearer, especially in the abstract and the end of the introduction.**

We agree with the reviewer's comment. Therefore, we modified the abstract and the introduction section, emphasizing more that the additional purpose of that work-apart from the development of a model that includes the fine,coarse and giant dust particles- is also to investigate the reasons behind the model overestimation of the dust particles' settling. The abstract section is given in lines 13, page 1 of the revised document:

*"Dust particles larger than 20 μm in diameter have been regularly observed to remain airborne during long-range transport. In this work, we modify the parameterization of the mineral dust cycle in the GOCART-AFWA dust scheme of WRFV4.2.1, to include also such coarse and giant particles, and we further discuss the underlying misrepresented physical mechanisms which hamper the model in reproducing adequately the transport of the coarse and giant mineral particles. The initial particle size distribution is constrained by observations over desert dust sources. Furthermore, the Stokes' drag coefficient has been updated to account realistic dust particles sizes ($Re < 10^5$). The new code was applied to simulate dust transport over Cape Verde in August 2015 (AER-D campaign). Model results are evaluated against airborne dust measurements and the CALIPSO-LIVAS pure dust product. The results show that the modelled lifetimes of the coarser particles are shorter than those observed. Several sensitivity runs are performed by reducing artificially the particles' settling velocities in order to compensate underrepresented mechanisms, such as the non-spherical aerodynamics, in the relevant parameterization schemes. Our simulations reveal that particles with diameters of 5.5-17 μm and 40-100 μm are better represented under the assumption of a 80% reduction in the settling velocity (UR80) while particles with sizes ranging between 17μm and 40 μm are better represented in a 60% reduction*

*in settling velocity (UR60) scenario. The overall statistical analysis indicates that the best agreement with airborne in-situ measurements downwind (Cape Verde) is achieved with 40% reduction in settling velocity (UR40). Moreover, the UR80 experiment improves the representation of the vertical structure of the dust layers as those are captured by the CALIPSO-LIVAS vertically-resolved pure dust observations. The current study highlights the necessity of upgrading the existing model parameterization schemes of the dust life-cycle components towards improving the assessment of the dust-related impacts within the Earth-Atmosphere system."*

Based on the reviewer's suggestion, we also modified the part of the introduction that summarizes the overarching goal of our study, the tools which have been utilized, and the justification and validity of our approach. The related part of the introduction is cited in line 95, page 4 of the revised document:

*"In this work, we demonstrate for the first time a method for incorporating coarse and giant desert dust particles (D > 20 μm, following the definition of the dust modes proposed in Ryder et. al, (2019), into the Advanced Research Weather version of the Weather Research and Forecasting (WRF-ARW) model in conjunction with the GOCART (Ginoux, 2003) aerosol model and the Air Force Weather Agency (AFWA) dust emission scheme (LeGrand et al., 2019) (WRF-GOCART-AFWA model). After pinpointing that the model quickly deposits coarse and giant dust particles, we investigate the reasons behind those findings: We use sophisticated in situ PSD measurements to initialize the model over the sources and to evaluate the simulated PSD over the receptor areas. We also use pure-dust spaceborne retrievals to assess the model performance in terms of reproducing the vertical structure of the dust layers. In addition, we perform a series of sensitivity tests by reducing the settling velocity of mineral particles in the model and we investigate the concomitant effects on dust fields. "*

▪ **I'm puzzled by the lengthy discussion of the inclusion of a new drag coefficient in section 2.1.2. I understand that a drag coefficient parameterization that is valid for larger Re number must be implemented since you're treating coarse and giant dust (with Re up to 10 or so), but I think the drag law you use (Eq. 14) is fairly standard. So rather than taking up the reader's finite attention with this lengthy description, I recommend you just state you implemented the drag coefficient law from Clift et al. (2005). Additionally, you should show that implementing this new**

**drag coefficient law is actually important by including a plot of the new and old drag coefficients versus particle size.**

One of the major and critical advancements in our work is the extension of the drag coefficient applicability for larger particles. This modification affects the simulated settling velocities and the particles' deposition rate. Despite the fact that the drag coefficient by Clift and Gauvin, 1971 is a standard drag coefficient, we have modified the way that the slip correction is applied compared to the WRF default version. In the WRF default code, the slip correction is applied unconditionally for any Re. However, in Mallios et al. (2020) it has been shown that the slip correction should be applied only in the Stokes regime Re<0.1. Thanks to this consideration (i.e., slip correction), a more realistic representation (i.e., large Re) of the settling velocities of larger particle sizes is achieved. Re is a function of size and settling velocity. The difference in drag coefficient is more significant for coarse and giant particles (approximately Re up to 10), where the drag coefficient given by (Clift and Gauvin, 1971) can be up to 2 times greater than that of Stokes. For this reason, we have decided to describe in detail how the new size-dependent drag coefficient is applied in our numerical experiments. We agree with the reviewer that in the initially submitted manuscript all these aspects were not clearly stated. Regardless, we have put an effort on reducing the length of the paragraph. The revised text (see lines 156-214, pages 6-8), is given below:

[revised manuscript text omitted]

"

Nevertheless, according to the reviewer's suggestion, we included a plot in the supplement, which shows a comparison between the drag coefficients given by Stokes and Clift and Gauvin (1971) along with its relevant difference with respect to the Reynolds number Re. The comparison in Fig. R1 shows that the drag coefficient by Clift and Gauvin (1971) can be 2 times higher than Stoke's drag coefficient, for Re up to 1, suggesting the necessity of the implementation of the revised drag coefficient. Figure R1 is also included in the Supplementary material (denoted as Fig.S1).

[Figure]

**Figure R1:** Drag coefficient for spheres given by the Stoke's approximation (black line) and the expression proposed by Clift and Gauvin (1971) (blue line). The red line represents the relative difference between the two drag coefficients.

Finally, we included a comment, related to the drag coefficient differences, in the revised document in line 204, page 7:

*"The departures between the drag coefficients given by Stokes theory and Clift and Gauvin (1971) become more evident for increasing particles' sizes. More specifically, the drag coefficient given by Clift and Gauvin (1971) can be up to 2 times higher than those of the Stokes' Law for coarse and giant particles (Fig. S1)."*

▪ **This paper was posted online a few days before the publication of a rather similar paper by Meng et al. in GRL that also found that the settling speed needs to be greatly reduced for a large-scale model to match measurements of coarse and giant dust particles. A brief comparison between the results in the two papers should be included.**

We totally agree. Therefore, a comparison between our results and those of Meng et al. 2022 is included in the Discussion section of the revised manuscript. We performed a short analysis of the differences in the settling velocities between the different scenarios hypothesized in the two surveys.

In our work, we reduced the settling by 20, 40, 60 and 80 per cent and found that a 60-80% reduction is needed to match the model with observations of giant and coarse particles in the vicinity of Cape Verde. In Meng et al. (2022), the authors, after reducing the settling velocity by 13% (UR13) for accounting for particles' asphericity based on Huang et al., (2020), performed sensitivity tests where they replace particle density (2500 kgm$^{-3}$) with lower values and found that a decrease in the modeled dust aerosol density by a factor of 10-20 ($\frac{\rho_D}{10}$ $or$ $\frac{\rho_D}{20}$), after accounting asphericity, is needed to improve the comparison between model and long-range dust observations of coarse particles. Despite the differences between the two works, especially in the calculation of the particles settling velocities (the two studies utilize different drag coefficients), the PSD which is used for the distribution of the emitted dust in the transport bins (the PSD in the Drakaki et al., (2022) parameterization is coarser for diameters greater than ~10 μm with respect to "observed FENNEC-PSD", Fig. R2) and the model spatial resolution (Meng et. al, (2022) uses the Comunity Earth System Model version 1.2 (CESM-v1.2) with the Community Atmosphere Model version 4.0 (CAM4) with 210 km x 277 km grid spacing, while in this study we use the WRF-Chem version 4.2.1 with the GOCART AFWA dust scheme with

15km x 15km grid spacing), both studies are suggesting a reduction in the settling velocity of the same order of magnitude. Fig. 5 shows a comparison of the different scenarios included in the two studies. The corresponding calculations have been performed assuming US Standard Atmosphere conditions. A reduction of particle density reduced by a factor of 10 (starting from the Clift and Gauvin (1971) drag coefficients) is almost equivalent to a decrease of 90% in the settling velocities. It is clear that a huge reduction in the settling velocity in both the Meng et al., (2022) methodology and this work is required, although the physical processes occurring to explain this reduction are not clear.

[Figure]

**Figure R2:** Comparison of the "fitted FENNEC-PSD" (red line) which is used in the dust parameterization in WRF-L and the PSD of the extended brittle fragmentation theory presented in Meng et. al., (2022) (yellow line). Red squares provide the "observed FENNEC-PSD" All volume size distributions are normalized in order to yield unity for the total volume of particles with diameters between 0.1 and 40 μm.

Figure R2 above shows the PSDs that have been utilized in the studies: Meng et al., (2022) in yellow and that in this study in blue. The black dashed line indicates the observed FENNEC-PSD (at 1km) as presented in Fig2a of the revised manuscript. All volume size

distributions are normalized in order to yield unity for the total volume of particles with diameters between 0.1 and 40 µm.

[Figure]

**Figure R3:** Terminal settling velocities with respect to particle diameter for dust particles, starting from the drag coefficient of Clift and Gauvin, (1971) and for the different scenarios described in Table R1.

Figure R3 above shows the terminal settling velocities with respect to particle diameter for dust particles, starting from the drag coefficient of Clift and Gauvin (1971) and for the different scenarios described in Table R1 below.

**Table R1: Different numerical experiments presented in Fig. S5**

| Cases | Description |
|---|---|
| UR60 | settling velocity reduced by 60% |
| UR80 | settling velocity reduced by 80% |
| UR13 | settling velocity reduced by 13% |
| ρD/10 | particle densities reduced by a factor of 10 |
| UR13&ρD/10 | particle densities reduced by a factor of 10 and settling velocity reduced by 13% |
| UR85 | settling velocity reduced by 85% |
| UR90 | settling velocity reduced by 90% |

Based on our reply we added a part in the Discussion section where we discuss the comparison between the two studies (see

lines 451-458, page 15). We also added Fig. R3 and Table R1 in the supplementary material, as Fig. S5 and Table S1 respectively.

*"Meng et al. (2022) performed a similar study, where after reducing the settling velocity by 13% for accounting for particles' asphericity based on Huang et al., (2020), they performed sensitivity tests reducing the dust particles' density from 2500 kg m⁻³ to 1000, 500, 250 and 125 kg m⁻³. They found that a decrease in the modelled dust aerosol density by 10-20 times its physical value (i.e., from 2500 kg m⁻³ to 250-125 kg m⁻3) is needed to improve the comparison between the model and the long-range dust observations of coarse particles. A 10 times reduction in particles' density is almost equal to a 90% reduction in the settling velocity (starting from the Clift and Gauvin (1971) drag coefficients and assuming conditions of U.S. Standard Atmosphere, Fig S5). It is clear that a huge reduction in the settling velocity in both the Meng et al., (2022) methodology and this work is required, although the physical processes occurring to explain this reduction are not clear."*

▪ **Lines 135-140 and Fig. 2: Here and elsewhere in the paper (section 2.2.2, Figure 5), not enough detail is provided on the used in situ measurements. Please describe exactly which runs were used for this data, how measurements were averaged over different runs and any other processing. Which instruments of the FENNEC and AER-D data did you use and how did you treat data that overlapped in the particle size range? And please include the measurement uncertainties and describe what's included in them.**

We agree with the reviewer that we have to clarify more the aspects relating to the measurements used in this work, the specific instrumentation and their corresponding errors. Regarding the PSD we use to modify the parameterization of emitted dust and calculate the dust fraction of emission for each size bin ($k\_factors$), we use size distributions from the Fennec field campaign from aircraft profiles over the Sahara (Mauritania and Mali) as described in Ryder et al. (2013). We select size distributions from"freshly uplifted dust" cases where dust is in the atmosphere for less than 12 h. Additionally, from these profiles we use data from the lowest available altitude, centered

[revised manuscript text omitted]

- **240-241: "The fine resolution increases the accuracy of the dust simulations and provides a good estimate of the missing mechanism." Please include either citations or original results that support this statement. Also, how does the fine resolution affect the numerical diffusion in the model? And please include a discussion in this section of the numerical diffusion in WRF-Chem as Ginoux (2003) hypothesized this to be a main factor in why coarse dust particles deposit too quickly in models. Currently, there's only a brief mention of this in the last paragraph of the paper but not really any discussion of how big a problem numerical diffusion is in WRF-Chem and thus of whether it can explain your results.**

The resolution applied here is adequate for the scale of phenomena we want to study. With the term "fine resolution" we wanted to denote that we have a finer resolution with respect to global datasets (e.g. 0.5 deg GFS), which will fail to reproduce the appropriate weather fields and dust fields (Cowie et al., 2015; Basart et al., 2016; Roberts et al., 2017; Solomos et al., 2018 ). However, the reviewer is correct that this can be misleading so we made changes in the original manuscript (lines 235-238, page 8).

*"The resolution applied in this study (15km grid spacing) is adequate for the scale of phenomena we want to study, improves the representation of topography and increases the accuracy of the reproduced weather and dust fields, compared to coarser resolution, such as used in global datasets (e.g. 0.5 deg GFS) (Cowie et al., 2015; Basart et al., 2016; Roberts et al., 2017; Solomos et al., 2018)."*

The WRF-Chem uses a spatially 5th-order horizontal advection and a 3-rd order vertical advection in the scalar conservation equation coupled with the 3-rd order Runge-Kutta time integration schemes which are non-diffusive schemes. Moreover, WRF-Chem uses the first order explicit advective scheme for the scalar concentration in the equation of gravitational settling. The first-order upstream scheme is notoriously too diffusive. Since the numerical diffusion is pointed out in Ginoux (2003) as a possible source of the model underestimation of dust coarse particles, the use of a less diffusive scheme in WRF settling parameterization could improve the accuracy of modelled dust concentration fields. Based on our reply we have added a discussion about numerical diffusion in WRF-Chem in Section 2.1.3 (page 238-244, line 8-9)

*"WRF-Chem solver uses a 5th-order horizontal advection scheme and a 3-rd order vertical advection scheme to solve the scalar conservation equation, along with the 3-rd order Runge-Kutta time integration scheme (Grell et al., 2005). The use of such high-order advective schemes eliminate the numerical errors of diffusion in the code. We should note though that in the deposition parameterization of GOCART-AFWA dust scheme the vertical advection of the losses due to the gravitational settling is solved by a first order explicit scheme, which is notoriously too diffusive (Versteeg and Malalasekera, 2007) and thus it can possibly induce numerical errors in the mass conservation (Ginoux, 2003)"*

- **Section 2.1.4: here the effect of asphericity on dust extinction is neglected, which could be substantial. I think that's fine as the focus is on the size distribution, but please note that simplification.**

Although we agree with the reviewer that the effect of the asphericity may be substantial, there is no available data (to our knowledge) of the extinction coefficient of dust particles with realistic irregular shapes. The commonly-used spheroidal shapes do not provide substantial differences for the extinction coefficient of the particles, at least when considering the aspect ratios measured for dust particles in Sahara (as these are provided by Kandler et al. (2009), as shown in Tsekeri et al. (2022). We have included the above in Section 2.1.4 in lines 255-273 and page 9:

*"Although the extinction coefficient values for spherical particles may be different from the extinction coefficient values of the dust particles, which have irregular shapes, to our knowledge there is no data available for the extinction coefficient of the latter. The extinction coefficient values of spheroidal shapes, commonly used as a proxy of the dust shapes, are not substantially different compared to the spherical particles (Tsekeri et al., 2022), at least when considering the aspect ratios measured for dust particles in Sahara (Kandler et al., 2009)."*

- **(16): here the units for dust mass concentration, particle density, and diameter don't match (they all use different length scales). Please correct.**

We would like to thank the reviewer for pointing this out. After the revision of the document the right equation is :

$$DOD_{550,n} = \sum_1^k \frac{3}{2\rho_{,k}D_{eff,k}} ML_{n,k}Q_{ext550,k},$$
(14)

Where $ML_{n,k}$ is the columnar dust load in g/m² for each grid box $n$ and for each model size bin $k$.

However, based on RC3 we removed the related part from Section 2.1.4.

- **Line 273: please elaborate on how you are "taking into account the absolute difference between WRF forecast time and Aqua overpass time"**

In the model, the DOD is computed in each grid model box and its instantaneous value is provided every one hour. The DOD value from Aqua satellite is acquired from the ModIs Dust AeroSol (MIDAS) DOD product, based on the following spatiotemporal collocation procedure: First, we reproject the model DODs on an equal lat-long grid at 0.4° x 0.4° spatial spacing. We should note that the model DOD field has no spatial gaps and is provided instantaneously for every hour. The MIDAS DOD is available in swath level (5-minute segments, viewing width of 2330 km) along the MODIS-Aqua polar orbit. Then, the two closest WRF outputs to the Aqua satellite overpass time are used to calculate a weighted-average WRF-DOD, by taking into account the temporal departure between forecast and overpass times, only for the WRF grid cell that coincides with the observations. Please note that we have removed the corresponding part related to b920 flight, based on RC3 comment.

**I find Figure 5 hard to interpret and I think a lot more information is needed here. The text notes (L. 347) that this result is for "an emission point in Mali" - could you indicate exactly what location? And are the model results here for the closest grid box? Did the model include emissions only from that grid box or from the entire domain? And see comments above on more details needed for the experimental data. Is this the same data as shown in Fig. 2a, except sorted into the five bins? And could you also include uncertainties on the measurements? I also recommend including your parameterized size distribution at emission to help interpret the model results.**

In Figure 5 we present the change of the PSD with height above an emission point in Mali, on 11/08/2015 at 14UTC. At this particular

time, a dust emission was initiated, with the maximum intensity for the broader area of Mali at the model grid point with altitude=24.9° and longitude=9.2°. The model PSDs in Figure 5 are from that grid box, after interpolating the model PSDs at 1, 2 and 3 km height. The red squares in Figure 5 correspond to the "observed FENNEC-PSD" (mean PSD of freshly uplifted dust cases at 1km) sorted into the five bins and the corresponding error bars indicate the maximum and minimum limits of the "observed FENNEC-PSD", sorted into the five model size bins, after including the standard deviation of "observed FENNEC-PSD". The black squares depict the "fitted FENNEC-PSD" sorted into five bins, which are used in the model parameterization to distribute the emitted dust mass into the five model bins. We agree with the reviewer that the description was incomplete and we have inserted additional information in the part of the discussion of Figure 5 in Sect 3.2 of the revised manuscript. Based on that and other reviewer comments we modify the Section 3.2 (see lines 334-359, page 12):

*"In Fig. 5 we present how the PSD varies with height above an emission point (latitude=24.9° and longitude=9.2°) in Mali, on 11/08/2015 at 14UTC. The model PSDs are only from that grid model box interpolated at 1, 2, and 3 km height and for the particular timestep (11/08/2015 at 14UTC). The red squares correspond to the "observed FENNEC-PSD" sorted into the five bins. The error bars provide the maximum and minimum limits of the "observed FENNEC-PSD", sorted into the five model size bins, after including the standard deviation of "observed FENNEC-PSD". The "observed FENNEC-PSD" (see Section 2.2.1) has been derived from several flights above dust sources, thus it is representative of the PSDs above Sahara sources and it used here as reference. The black squares depict the "fitted FENNEC-PSD" sorted into five bins, used in the model parameterization to calculate the emitted dust mass of the corresponding five model transport bins. The difference between the "fitted FENNEC-PSD" and the "fitted FENNEC-PSD" occurs due to the fitting process. The modelled volume concentration is reduced with height by an order of magnitude between 2 and 3 km for particles with diameters 17-40 μm (bin 4). At 3km the simulated concentrations of particles in bin 4 and bin 5 are very low compared to the measurements in Fig. S2a of Ryder et al., (2013a) which indicate the removal of giant particles above 4 km (Ryder et al., 2013a, Figure S2a). Although a direct comparison between the modelled and the observed PSD for this particular emission point is not feasible, since the FENNEC campaign took place on different dates than the AER-D and there are no available measurements above dust sources for the period we performed our simulations, we note a modification of the PSD shape, both*

*for model and observations at 1km. It is evident that the model overestimates the PSD for bins 1-3 while the opposite is found in the size spectrum of the super-coarse (bin4) and giant (bin5) dust particles. Therefore, a model weakness is revealed at the very early phase of the dust transport. Those differences can be attributed to an overestimation of their loss during uplift from the surface to 1 km, or to higher updrafts that remain unresolved in our numerical experiment. Another possible source of this underestimation could be the utilization of a not well-defined PSD shape constraining the distribution of emitted dust mass to the model transport size bins. The use of a PSD with a higher contribution of coarse and giant dust particles could possibly improve the representation of the coarse and giant particles aloft (Fig. S2 and S3) and can be assessed in future studies. Additionally, comparing the "observed FENNEC-PSD" with the modelled PSD of the scenario with the maximum relative reduction of the settling velocities (UR80) in Fig. 5, we find a significant increase of the modelled volume concentrations, reducing the differences seen in volume concentrations in bin4 and bin5 without the reduction of the settling velocity, although the underestimation in bin 5 is still evident."*

Moreover, we plot again Figure 5 after other reviewers' recommendations, adding the observed mean PSD of "freshly uplifted dust" during Fennec 2011 in red squares ("observed FENNEC-PSD sorted in 5 bins") along with the standard deviation of the observed values within the size bin. The "fitted FENNEC-PSD sorted in 5 bins", which is used at the parameterization of the emission, is depicted with black squares in the revised plot. Here we must note that the label in the legend "observations" was not correct, since, in Figure 5 of the original, the red squares corresponded to the "fitted FENNEC-PSD" instead. We corrected the legend accordingly. The revised figure is inserted in the revised document in lines 915-922 and page 34

"

[Figure]

*Figure 5: Dust size distribution above an emission model grid point (latitude=24.9º and longitude=9.2º) in Mali, on 11/08/2015 at 14UTC. Blue solid line: the modelled dust PSD of the CONTROL run interpolated at 1 km altitude above the dust source, orange solid line: the modelled dust PSD of the CONTROL run interpolated at 2 km altitude above dust source, green solid line: the modelled dust PSD of the CONTROL run interpolated at 3 km altitude above dust source, blue dotted line: the modelled dust PSD of the UR80 run interpolated at 1 km altitude above the dust source and red squares: the "observed FENNEC-PSD" sorted in 5 bins (observed at 1 km altitude), black squares: the "fitted FENNEC-PSD" sorted in 5 bins which has been used for the distribution of the model emission to the five size bins used in the model."*

- **L377-380: Why do you average over the eight neighbouring grid points when you're already interpolating the measurements? Some more explanation is needed here.**

We agree with the reviewer that we should explain more the way that model results are collocated to the AER-D measurements. In the collocation procedure, we firstly interpolate the model dust fields to the specific height of each flight leg (or "RUN"). Afterwards, we find the closest grid box to the flight leg coordinates and its eight neighbouring grid boxes in the same level height. The selection of the model grid points is done from the two hourly model outputs that enclose the time of flight RUN. Finally, the dust field is averaged among the selected grid boxes and selected times and the variability is expressed in terms of the standard deviation. In case that the flight time coincides with the model output, we include in the averaging that particular hourly model output as well as the next and previous hours' outputs. Based on our reply, we updated the related parts in the revised manuscript in lines 374-383, page 13:

*"The red squares correspond to the observations and the error bars denote the total (random and systematic) measurement error (see Sect 2.2.1). The modelled PSDs are collocated in space and time with the measurements of each flight segment. For each flight segment, we extract the modeled PSD by interpolating the dust field to the specific altitude of the flight RUN. Additionally, we average the dust field of the nearest grid cell to each coordinate pair along the flight segment track, and the eight neighbouring grid cells of the same altitude. The coordinates of the flight leg track are depicted with orange dots and the collocated grid points used for deriving the modelled PSD (at the specific height of each flight leg) with blue dots. In the time dimension, we average the two hourly model outputs that contain the times of the measurement. In case that the time of measurement coincides with the exact hourly output, the model output on that hour along with the outputs prior and after that are averaged. The error bars in the model PSDs indicate the standard deviation of the collocated grid points averaging in space and time."*

- **Figure 8: Please describe what exactly the error bars represent. Is this derived from the counting uncertainty in a given run? Or the standard deviation (or standard error?) over several measurements?**

We agree with the reviewer that we have to provide more information about the uncertainties of the measurements. The error bars in Fig. 7 (Fig 8 in the original manuscript) correspond to the total

measurement errors (random and systematic) due to the counting error, the discretization error, the uncertainties in the sample area and the uncertainties in the bin size due to Mie singularities (Ryder et al., 2018). Therefore, we modified the revised manuscript including a description in the caption of Figure 7 in lines 928-933, p 36:

"*Figure 7: Modeled and observed dust PSD of flight b928, over the Atlantic Ocean during AER-D, for straight-level-runs (a) R02, (b) R03, (c) R05, (d) R06, (e) R10, (f) R11 and (g) R12. The in situ observations are shown with red squares (along with the total measurement error). The collocated modeled PSDs are shown with lines, for the CONTROL run (black), UR20 (blue), UR40 (orange), UR60 (green), and UR80 (purple) and the corresponding standard deviation with the associated error bars. The brown vertical lines indicate the limits of the model size bins. The inlet maps show the flight segment track and the collocated model grid points.*"

We also modified Section 2.2.1 by adding the description of the error bars on the revised manuscript (page 10-11, lines 303-305):

*"The error bars represent the total (random and systematic) measurement error due to the counting error, the discretization error, the uncertainties in the sample area and the uncertainties in the bin size due to Mie singularites (Ryder et al., 2018)."*

We also modified Section 3.4 by adding the description of the error bars on the revised manuscript (page 13, lines 374-375):

*"The red squares represent the observations and the error bars represent the total (random and systematic) measurement error (see Sect 2.2.1)."*

- **Also for Figure 8: I find the results in Fig. 8a puzzling. The measurements shown here are at the very lowest level, only 38m above the ground. So presumably, these measurements were part of the data used in Fig. 2 to parameterize the emitted size distribution, is that correct? Then why does the model do so poorly in reproducing these measurements so close to the surface? Please show the emitted size distribution in this plot to help the reader interpret your model results. Please also discuss why the model does not capture the measurements so close to the**

**ground, where errors in deposition would presumably have not as much impact on the results.**

The observations depicted in Figure 7 (Figure 8 in the original manuscript) have been derived from the AER-D campaign, in the downwind area, over the Eastern Atlantic Ocean far from the dust sources. Thus, those observations contain the effects of dust deposition and dust transport. None AER-D data have been used for the model parameterization (only data from the FENNEC campaign have been used for the model parameterization, as described in Sections 2.1.1). The fact that the model underestimates the presence of the coarse and giant particles confirms that there is one or more physical mechanisms of the dust transport that the model misses or underrepresents. Obviously, this is not clear enough in the original manuscript, thus we add some text lines in Section 3.4 of the revised document (lines 370-374, p.13 ) noticing that issue:

"Figure 7 illustrates the simulated PSDs from each experiment (i.e., CONTROL and URx), along with those acquired by the airborne in situ measurements at different segments and altitudes of the flight b928 in the surrounding area of Cape Verde (downwind region). For the other AER-D flights (i.e., b920, b924, b932 and b934) similar findings are drawn and for brevity reasons are omitted here and are included in the supplementary material (Fig.S4). All AER-D measurements demonstrate the impacts of the processes that are associated with dust transport."

▪ **Figure 10: What are the gray, yellow, and blue shading here?**

The shaded areas correspond to altitude ranges within the atmosphere. In order to simplify the plots in Figure 9a (Figure 10a in the original manuscript) and make both plots (a and b) more consistent with each other, we choose to remove the shading from Figure 9a. Below is the revised Figure 10a:

[Figure]

Also, in the revised Fig. 9 (Figure 10 in the original manuscript), based on other reviewers' suggestion, we included a figure showing the daytime and nighttime overpasses of CALIPSO. The revised Fig.9 is inserted in lines 947-955, page 38 of the revised manuscript:

[Figure]

*"Figure 9: (a) Profile of the mean extinction coefficient at 532 nm, by LIVAS pure-dust product (black red line), and profiles of the mean extinction coefficient at 532 nm simulated from the different experiments of Table 3 (CONTROL, UR20/40/60/80). The orange shading indicates the standard deviation of the LIVAS profile averaging. (b) The mean absolute biases between the LIVAS profile and the simulated profiles from the different experiments, in the domain of interest, between 05/08/2015 and 25/08/2015. The vertical dashed lines are the mean absolute bias between the LIVAS profile and the simulated profiles from the different experiments averaged over the altitudes of region II. (c) The domain of interest and the daytime (red) and nighttime (blue) CALIPSO overpasses. The vertical dashed lines are the mean absolute bias between the LIVAS profile and the simulated profiles from the different experiments averaged over the altitudes of region II."*

▪ **Discussion and conclusion section: As written, this is really only a discussion section. I recommend the authors add a summary of the results of their study for the reader.**

We would like to thank the reviewer for the suggestion which will improve the structure and the presentation of our work. Below is the part of the Summary and conclusions in lines 504-546, page 17-18:

[revised manuscript text omitted]

- **441: The gravitational force acts on the center of mass and thus does not create a torque. Perhaps you mean that the aerodynamic force creates a torque? Please correct.**

We would like to thank the reviewer for pointing this out. Indeed, we meant the aerodynamic force instead of the gravitational force. Please note that based on that and RC3 comments we have revised the Section of the discussion and the referring part is omitted.

- **438-455: This is an interesting discussion of the effects of shape and particle orientation on settling speed. It left me confused on a few points though. The text states that "prolate spheroids fall faster than their spherical counterparts" even though their surface area is larger. How is that possible as more surface area would create more drag? This conclusion is also opposite of results in, for instance, Ginoux (2003). Do you perhaps mean that for this statement to apply to the special case when the prolate spheroid is aligned with its longest axis in the vertical direction, such that its cross-sectional area is smallest? If**

**not, wouldn't the drag of the spheroid relative to an equal-volume sphere depend on the orientation, which itself is unknown as it depends on a variety of factors including the electric field (per Mallios et al. 2021)?**

Obviously, the statement here needs more explanation to avoid any misunderstanding. In lines 445-448 of the original manuscript, we state that "prolate spheroids fall faster than their spherical counterparts of the same volume" without specifying the orientation because prolate spheroids fall faster than their spherical counterparts of the same volume regardless of particle's orientation (Mallios et al., 2020). Then in the next sentence, we explain that this happens due to two reasons. One is the projected area (and not the surface area) that depends on the particle orientation, and the other is the drag coefficient which is shape and orientation dependent. We would like to emphasize that the resultant drag force is proportional to the product of projected area and drag coefficient. Mallios et al. (2020) have shown clearly, that in the case of spheres and prolate spheroids (in the aspect ratio range 1.4-2.4) of the same volume, prolate spheroids fall faster, because the product of drag coefficient time the projected area is always smaller in the case of the prolate spheroids.

Moreover, we would like to add that the use of the surface area of the particle in the interpretation of the drag force's behavior can be misleading in that case and should be avoided. A prolate spheroid with a given major axis and aspect ratio has a specific value of surface area, regardless of its orientation. On the other hand, the projected area of the particle changes with orientation, affecting the drag force and the particle's settling velocity.

Finally, we would like to add that the statement "prolate spheroids fall faster than their spherical counterparts of the same volume" does not contradict the findings of Ginoux (2003), because they compared

prolate spheroids and spheres of the same cross section (in their Eq. 10 the equivalent diameter Dp is calculated as the diameter of a sphere with the same cross section of a randomly oriented prolate spheroid). For a more detailed explanation of this, please check our response to the reviewer's next comment.

- **Later in this same section you seem to state the opposite conclusion (L. 452-5), that prolate spheroids do fall slower than spheres. But I think here the difference is that you're comparing it to spheres of the same max dimension (rather than volume)? I think this is quite confusing to the reader and I recommend you focus on the comparison that could actually explain that particles settle slower than your model simulations predict. And these measurements are presumably for volume-equivalent spheres?   Or are these optical diameters, so it depends on the particle index of refraction and the shape of real dust particles? That should also be discussed in section 2.2.1 for the discussion here to add value. In general, I think the discussion on the effects of asphericity on settling should be presented more clearly for the statement on L. 476 ("the particle asphericity seems to be a strong candidate for the suggested corrections") to make sense to the reader.**

We can understand the reviewer's confusion. Even today, there is not a definite answer to the question "which one does fall faster? A sphere or a prolate spheroid?". The reason is that there are many parameters that influence the comparison. Ginoux (2003) compared randomly-oriented prolate spheroids and spheres of the same cross section. They showed that although spheroids fall slower, the difference between spheres and spheroids is negligible for aspect ratio values less than 5.

Huang et al. (2020) compared randomly-oriented ellipsoids and spheres of the same volume. They showed that ellipsoids fall around 20% slower than spheres.

Mallios et al. (2020) compared prolate spheroids and spheres of the same maximum dimension, and of the same volume. Moreover they did not assume randomly-oriented particles, but particles of specific orientation (horizontal and vertical). They showed that the results of the comparison change when the maximum dimension or the volume-equivalent size is used in the comparison changes (maximum dimension or volume). Prolate spheroids fall slower than spheres of the same maximum dimension, regardless of orientation. On the other hand, prolate spheroids fall faster than spheres of the same volume, regardless of orientation. The comparison with in situ observations of the maximum dimension of particles is not so common, since most of the in situ measurements do not provide the sizing of the particles in terms of their maximum dimension, with some exceptions, as e.g. the observations shown in van der Does et al. (2016) of individual giant mineral particles (larger than 100 μm in maximum dimension).

Based on our reply to this and the previous reviewer's comment, we modified the effects of asphericity discussion in page 15-16, lines 461-480 of the revised document as:

*"One of the processes proposed in the literature to explain the longer atmospheric lifetimes of large mineral dust particles is the particle asphericity. Ginoux (2003) compared randomly-oriented prolate spheroids and spheres of the same cross section. They showed that spheroids fall slightly slower than their spherical counterparts, with their difference being negligible for spheroids with aspect ratio values less than 5.*

*Huang et al. (2020) compared randomly-oriented ellipsoids and spheres of the same volume. They showed that ellipsoids fall around 20% slower than spheres.*

*Mallios et al. (2020) compared prolate spheroids and spheres of the same maximum dimension, and of the same volume. Moreover, they did not assume randomly-oriented particles, but particles of specific orientation (horizontal and vertical). They showed that the results of the comparison change when the maximum dimension or the volume-equivalent size is used in the comparison. Prolate spheroids, with*

*aspect ratio values in the range of 1.4-2.4, fall slower than spheres of the same maximum dimension, regardless of orientation, with the relative difference between the settling velocities reaching the value of 52%. On the other hand, prolate spheroids, in the same aspect ratio value range, fall faster than spheres of the same volume, regardless of orientation. The comparison with in situ observations of the maximum dimension of particles is not so common, since most of the in-situ measurements do not provide the sizing of the particles in terms of their maximum dimension, with some exceptions, as e.g. the observations shown in van der Does et al. (2016) of individual giant mineral particles (larger than 100 μm in maximum dimension).*

*All the above show that more work is needed for the definite and accurate quantification of the particle asphericity effect on their settling. Nevertheless, there are indications pointing that aspherical particles remain in the atmosphere longer, and that asphericity can be one of the reasons for the differences between the modelling results and the observations."*

**I think the author contribution sections require more detail. There are a large number of authors with only a generic description of their contributions, with only the descriptions for ED, VA, AT, EP, and AG more specific. I think the contributions of each individual author should probably be spelled out more.**

Absolutely. We revised the related part in the revised document (line549-564, page18-19) as:

*"**Author Contributions:** ED, VA, and AT design the study; SM guided ED on the methodology for the replacement of the drag coefficient. AT provided useful assistance on the treatment of airborne observations. CR provided the data from the airborne in situ measurements and provided useful information about the instrumentation methods; ED developed the code, performed the simulations and analyzed the results. AG and CR consulted ED on the methodology of in situ and WRF datasets. VA, EM and EP provided the LIVAS dataset, lead the collocation methodology and helped on the interpretation of the results. ED plotted the model and observation data (apart from LIVAS). EP treated and plotted LIVAS data; ED wrote the manuscript draft; VA, AT, AG, EP, SM, CS, SS, EM, CR, DB and PK provided critical feedback and reviewed and edited the manuscript."*

**Technical corrections:**

- **Can you provide a reference for Eq. 10?**

Absolutely. In the revised manuscript Eq.10 became Eq.9. So, Eq.9 of the revised manuscript is an equation of dynamic viscosity μ, based on the kinetic theory and comes from the general expression of Sutherland's Law:

$$\mu = \mu_o \cdot \frac{\left(\frac{T}{T_o}\right)^{3/2} (T_o + S)}{(T + S)}$$

but its constants are based on experiments (United States Committee on Extension to the Standard Atmosphere., 1976;). $S$ is the Sutherlands constant equals to $110.4\,K$ and $\beta$ is a constant equals to $1.458 \cdot 10^{-6}\,kg \cdot m^{-1} \cdot s^{-1} \cdot K^{-1/2}$. The value of β corresponds to a reference air temperature (To) of 273.16 K and an air viscosity (μo) at To which is equal to $1.716 \cdot 10^{-5} kg \cdot m^{-1} \cdot s^{-1}$ (White, 2006; Hilsenrath, 1955). We should note that this equation is included also in the original parameterization of AFWA-GOCART in WRF model. Based on our reply we provided the references for Eq.9 according to the reviewer suggestion in page 7 and line 187-193

[revised manuscript text omitted]

---

## Author Comment (AC3)

**The authors apply the WRF-Chem model to simulate coarse and giant dust (besides the fine dust). For this purpose, they modified the dust transport bins in WRF-Chem, applied a modified (observations-derived) pre-defined particle-size distribution (PSD) to dust at emission and also modified the settling velocity to be applicable beyond the Stokes regime. With their modified model version, they conduct sensitivity runs with reduced settling velocities to test the impact of settling velocity compared to aircraft dust observations.**

**The study is timely and interesting, but I see two major weaknesses, one related to the comparison with observations and the other related with the transfer of the simulation results to processes other than settling velocity. I detail those aspects below besides other specific comments.**

**While the manuscript is overall well organised (although I suggest some changes, see below), grammar and orthography need to be improved throughout the manuscript.**

We would like to thank the reviewer for taking the necessary time and effort to review our manuscript. We sincerely appreciate all your valuable comments and suggestions, which helped us in improving its quality. We have put a lot of effort on improving grammar and orthography issues in the revised manuscript and we tried to incorporate all the changes to reflect the provided suggestions.

**Main comments:**

**The authors distribute the total emitted dust mass (all sizes) across their new bins using a prescribed PSD obtained from aircraft observations (FENNEC-PSD) at 1 km altitude. Given that particles**

settle when they are airborne (even if less than expected), the actual PSD at emission has to have been coarser than that observed at 1 km. It is still possible technically and no issue to apply this observed PSD at 1km to the emissions. However, in Fig. 5 / Section 3.2 / Section 3.4 / Discussion, the authors compare the modelled PSD at 1, 2, and 3 km height with the mean FENNEC PSD at 1km height and conclude that the model underestimates coarse dust, even when the settling velocity is reduced by 80 %. This is only natural as the FENNEC-PSD has been used at the PSD at emission, hence the model could only ever reproduce the observed PSD at 1km if all the emitted dust would be transported to 1 km in the model without any sedimentation.  the model has no chance to do so. If this is the goal, then the PSD at emission would need to be described as coarser than the FENNEC-PSD, possibly by assuming a certain settling rate. In the context of the comparison between modelled PSDs and those observed in AER-D, I would like to see a specific comparison between the AER-D PSDs, the FENNEC-PSDs (this could in principle be seen from Figures in the paper, but a direct comparison would make this much easier): Are those PSDs, which have been measured above (FENNEC) and distant (AER-D) to dust source regions "sufficiently" distinct (i.e. is the FENNEC-PSD, which has been used for the emission, "sufficiently" [whatever this means] coarser than the AER-D PSD), such that the model has a chance to reproduce the after-transport AER-D PSD? I believe this aspect is critical, because it might well be that settling is one, but not the only key problem, but that particle sizes at emission are considerably underestimated, even if using the FENNEC-PSD. Besides this general discussion, I would like to ask how the part of the FENNED-PSD, that extends beyond 100 microns, has been dealt with when distributing the emissions. Was this fraction ignored and the remaining PSD re-normalized?

The reviewer raises an important issue on the impact of the size distribution which is used for the parameterization. Let us first mention that the critical information that passes into the model, and it is derived from the "fitted FENNEC PSD", is the PSD shape and not its exact magnitude. We use only the part of FENNEC-PSD between 0.2 to 100µm in diameter, which is normalized, after ignoring the remaining part that extends below 0.2 µm and beyond 100µm.

With regard to the question about whether the FENNEC PSDs are significantly different to those from AER-D, in order that the model has a chance to demonstrate the appropriate downwind changes in dust PSD: A comparison of the PSDs between FENNEC and AER-D is given in Ryder et al. (2019) Figure 2. There, the differences between the FENNEC and AER-D are around a factor of 10 at 20 microns diameter, and nearly a factor of 100 at 60 microns diameter. Note that this figure shows the Fennec mean, rather than the low-altitude fresh PSD used in our study, which contains an even stronger contribution from the coarse and giant size ranges, such that the differences will be even greater. Therefore, there is ample scope for the model to demonstrate an (in)ability to transport and alter the full size distribution.

Nevertheless, it is quite difficult to quantify exactly the differences in the shape of the PSD at 1 km and the PSD near the surface. This is the reason why we choose to rely on the measured FENNEC-PSD at 1 km neglecting the sedimentation from this altitude down to the surface. However, following the reviewer's recommendation, we calculated a settling rate from the volume difference with height (dV/dz), derived from the "observed FENNEC-PSD" of freshly uplifted dust at height 1 and 1.5 km (Figure S2a from Ryder et al., 2013). Given that settling rate, we estimated with extrapolation, a new volume size distribution, called "FENNEC-PSD-0km", which corresponds to the surface (blue squares, Figure R1a). The blue solid line corresponds to the lognormal size distribution fitted to the blue squares, hereafter "fitted FENNEC-PSD-0km". Then, we calculated the new

$k_{factors}$ based on the volume size distribution "fitted FENNEC-PSD-0km". A comparison between the previously used $k_{factors}$ (based on the "fitted FENNEC-PSD", black line) and the new ones (based on the "fitted FENNEC-PSD-0km", blue line) is shown below in the Figure R1b: It is evident that the contribution of bin 5 is greater for the "fitted FENNEC-PSD-0km" than for the "fitted FENNEC-PSD" (at 1km), which is used in the paper. We use the new $k_{factors}$ to run an additional simulation CONTROL-nPSD. A comparison of the volume size distribution above an emission grid point (similar to Figure 5 in the manuscript) from CONTROL (blue line) and CONTROL-nPSD (orange line) runs is presented in Figure R2. According to the results we notice only a small improvement using the fitted FENNEC-PSD-0km. Despite the small improvement, the results suggest that the use of a coarser PSD has the tendency to improve the model PSD representation. Possible new measurements from near the sources, which reveal coarser PSDs near the ground, could be used in future studies.

[Figure]

[Figure]

**Figure R1:** a) "fitted FENNEC-PSD" (black line), FENNEC-PSD-0km (blue squares) estimated with extrapolation, applying the same volume difference per meter as that which holds between the measurements at 1km and 1.5km , and the corresponding "fitted FENNEC-PSD-0km" (blue solid line).

[Figure]

**Figure R2:** The "observed FENNEC-PSD" of freshly uplifted dust during FENNEC 2011 campaign (red squares), the model volume PSD above a source grid point from CONTROL run (blue line) and from CONTROL-nPSD run (orange line).

Based on our reply we included Fig R1 and Fig R2 in the Supplementary material (as Fig. S2 and Fig. S3, respectively) and added a related comment in the discussion of Fig. 5 of the revised manuscript, regarding the model underestimation for bin 4 and 5 above dust sources in lines 350-355, page 12:

*"Therefore, a model weakness is revealed at the very early phase of dust transport. Those differences can be attributed to an overestimation of their loss during uplift from the surface to 1 km, or to higher updrafts that remain unresolved in our numerical experiment. Another possible source of this underestimation could be the utilization of a not well-defined PSD shape constraining the distribution of emitted dust mass to the model transport size bins. A use of a PSD with higher contribution of coarse and giant dust particles could possibly*

*improve the representation of the coarse and giant particles aloft (Fig. S2 and S3) and can be assessed in future studies.”*

**I understand that the sensitivity experiments on settling have been performed to mimic the effects of other processes. This is particularly applicable to effects of particle asphericity. However, the effects of other processes mentioned in the introduction, e.g. turbulence or vertical mixing in the Saharan Air Layer, are most likely much less homogeneous than settling and much more closely related to the meteorological conditions. I am not convinced that sensitivity experiments on the settling velocity are suitable to represent the effects of these processes. My recommendation is therefore to focus the manuscript on settling (which contains uncertainties as well, e.g. due to asphericity) and only discuss the other processes as possible additional contributors.**

We agree with the reviewer's concerns, thus we have revised the related parts of the manuscript according to reviewer's recommendation in Section 2.1.3 (see lines 244-254, page 9):

*"A series of additional sensitivity runs has been performed aiming to resemble possible mechanisms (misrepresented or even absent in the model) counteracting gravitational settling towards reducing the differences between the CONTROL run calculations and the in-situ observations (shown in Sect. 3.4). To be more specific, we gradually reduced (with an incremental step of 20%) the settling velocity by up to 80%, with the corresponding runs named as URx (x corresponds to the reduction in percentage terms). Under such theoretical conditions, it is expected that the giant dust particles will be suspended for longer periods and that they will be transported at larger distances than the current state-of-the-art models simulate, failing to reproduce what is observed in the real world. Based on these sensitivity experiments, we defined a constant (by percentage) relevant reduction of the particles' settling, which in its absolute value varies with size. Therefore, it is more similar to the effects that are related to aerodynamic forces due to the non-spherical shape and the orientation of the suspended dust particles (Ginoux, 2003b; Loth, 2008; Zastawny et al., 2012; Shao et al., 2017; Sanjeevi et al., 2018; Mallios et al., 2020)."*

**At several locations in the manuscript average PSDs or other quantities are discussed, but (some examples are mentioned below), but it was often not clear to me what averages those are**

**(temporal, spatial, weighted?). I might have missed it, but I suggest to check this and clearly state how the shown and discussed quantities have been calculated.**

We agree with the reviewer, and after also taking into account the other reviewers' comments, we have revised the manuscript providing more details regarding the modelled and observed PSDs in several parts of the revised manuscript.

In Section 2.1.1, we describe the methodology for the modification of the dust parameterization in WRF-L and we refer to the PSD data that we have used to do so, in line 141-148, page 5:

*"We rely on prescribed PSD for the emitted dust particles at the source based on the airborne in situ measurements acquired during the FENNEC campaign of 2011 (Ryder et al., 2013a). More specifically, for the freshly uplifted dust we use the mean PSD at the lowest available height (i.e., 1km) t, obtained by averaging profile measurements above the Sahara (Mauritania and Mali), hereafter called the "observed FENNEC-PSD", which is shown in Fig. 2(a) with red squares. Figure 2a shows also the "fitted FENNEC-PSD" (solid red line), which is the fit of the "observed FENNEC-PSD", using five lognormal modes (Table 4). In Sect. 2.2.1 more information is provided on the derivation of the mean "observed FENNEC-PSD", including also the description of the FENNEC 2011 campaign, the in-situ instrumentation used and the processing of the acquired data."*

We also modified the caption of Fig.2 which presents the data which are used for the parameterization, by providing more details in line 899-905, page 31 of the revised manuscript:

*"Figure 2: Prescribed dust size distribution used in the WRF-L for the distribution of total dust mass to the transport model size bins: (a) "observed FENNEC-PSD" ($\mu m^3 cm^{-3}$) (red squares), and the respective "fitted FENNEC-PSD" (red solid line). The "observed FENNEC-PSD" corresponds to the PSD observations at 1km, obtained by averaging profile measured data of freshly uplifted dust cases, over 500m. The shaded areas indicate the model transport size bins in WRF-L. Error bars indicate the standard deviation of the observed values (b) The $k_{factors}$ of the transport size bins calculated based on "fitted FENNEC-PSD", provide the mass fraction of the emitted dust for each bin."*

We also modified Section 2.2.1, where we describe in a more detailed way the PSDs that we have used in this work, both for the modification of the dust parameterization and the evaluation of the model results. In the

additional information, details regarding the instrumentations and the respective measurement uncertainties are included. The revised section is positioned in line 278-306, page 10-11:

*"During the FENNEC field campaign in 2011 (Ryder et al., 2013b, 2013a) and the AER-D field campaign in 2015 (Ryder et al., 2018, 2019), airborne in situ observations were collected with the FAAM BAE research aircraft.*

*In this study we use size distributions from the FENNEC field campaign, aquired during aircraft profiles over the Sahara (Mauritania and Mali), as described in Ryder et al. (2013a). We select size distributions from "freshly uplifted dust" cases, when dust particles are in the atmosphere for less than 12 h. Additionally, from these profiles we use data from the lowest available altitude, centered at 1km, covering altitudes between 0.75 to 1.25km. The derived PSD is depicted in Fig.2(a), hereafter referred to as the "observed FENNEC-PSD". Error bars in Fig.2(a) indicate the standard deviation of the observed values across the profiles and altitudes we used. The instrumentation for those measurements was the Passive Cavity Aerosol Spectrometer Probe (PCASP, 0.13-3.5 μm), the Cloud Droplet Probe (CDP, 2.9-44.6 μm), using light scattering measurements and assuming a refractive index (RI) of 1.53-0.001i (which is constant with particle size), spherical shape for the particles, and using Mie calculations to convert from optical to geometric diameter, as well as the Cloud Imaging Probe (CIP15, 37.5-300 μm)). The instruments and data processing are described in Ryder et al. (2013a). The midpoint size bin diameters do not overlap, though there is some overlap in bin edges between the instruments. A fit on the observations is provided in Figure 2a (the "fitted FENNEC-PSD" with solid red line), which is used in the parameterization of the emitted dust, as described in Section 2.1.1, to modify the GOCART-AFWA dust scheme in WRF.*

*We also use PSD observations during horizontal flight legs at a constant height (referred either as RUNs or flight segments) over the Atlantic Ocean during AER-D. We use measurements taken with PCASP (D =0.12-3.02 μm) for fine dust particles. For the coarse and giant mode of dust we used measurements from CDP (D=3.4-20 μm, although CDP measurements availability extends up to 95.5 μm as it is explained below) and the two-dimension Stereo probe (2DS, D = 10–100 μm -although the instrument measures up to 1280 μm few particles larger than 100μm were detected). For the light scattering techniques of PCASP and CDP, a RI = 1.53-0.001i is assumed for the conversion of the optical to geometric diameter (as in FENNEC 2011 campaign). CDP observations extend up to the size of 95.5 μm, thus data from CDP and 2DS partly overlap in their size range. Since 2DS observations are more reliable in the overlapping size range, we used the CDP observations for particles with sizes up to 20 μm. Also, 2DS-XY observations are preferred over the 2DS-CC, since they better represent the non-spherical particles. A more detailed description of the in-situ instruments and the corresponding processing of the data acquired during the AER-D campaign is included in Ryder et al., (2018). The error bars represent the total (random and systematic) measurement error due to the counting error, the discretization error, the uncertainties in the sample area and the uncertainties in the bin size due to Mie singularites (Ryder et al., 2018). All PSD measurements are at ambient atmospheric conditions. The locations of the flights of AER-D used in this study are depicted in Fig.3."*

In Section 3.4 of the revised document, we provided more information about the collocation of the model data to the AER-D flights in line 375-383, page 13:

*"The modelled PSDs are collocated in space and time with the measurements of each flight segment. For each flight segment, we extract the modeled PSD by interpolating the dust field to the specific altitude of the flight RUN. Additionally, we average the dust field of the nearest grid cell to each coordinate pair along the flight segment track, and the eight neighbouring grid cells of the same altitude. The coordinates of the flight leg track are depicted with orange dots and the collocated grid points used for deriving the modelled PSD (at the specific height of each flight leg) with blue dots. In the time dimension, we average the two hourly model outputs that contain the times of the measurement. In case that the time of measurement coincides with the exact hourly output, the model output on that hour along with the outputs prior and after that are averaged. The error bars in the model PSDs indicate the standard deviation of the collocated grid points averaging in space and time."*

Additionally, we revised Fig.3, describing in more detail, the AER-D campaigns flight segments that we have used in this work, their locations and the respective collocated model grid points that we used for the model-observation comparison in Fig.8 and Fig S4. The revised Fig. 3 is inserted in lines 896-900, page 32 of the revised manuscript:

[Figure]

*Figure 3: Domain and topography map of the WRF-L model simulations, with a horizontal grid spacing of 15km, and 70 vertical levels. The tracks of the AER-D flights, used in this study (b920, b924, b928, b932 and b934), are depicted in the central plot with different colours. The aircraft tracks of each flight RUN are also*

*depicted with the orange dots in the surrounding maps. The blue dots correspond to the collocated model grid points."*

In Section 3.3 we discuss Fig.6 which shows the mean modelled dust load temporarily averaged of the CONTROL experiment, averaged over the period of the simulation (5/8/15- 25/8/15) after neglecting the period that is used as the simulation spin-up. We modified the related part in page 12, lines 361-362 to avoid any misunderstanding:

*"In Fig. 6, the spatial patterns of the columnar dust concentrations are depicted, averaged over the period of 5/8/15- 25/8/15, for the total mass as well as for each one of the five size bins simulated with the CONTROL run."*

We also revised Section 3.4, where we discuss Fig.8 (Fig.9 in the original manuscript). In Fig.8a we present the mean modelled and observed PSDs during the AER-D within all flight segments of Fig.3 (including the neighbouring points). Since the location of each flight segment and the respective time of measurement is different (the exact locations are presented in the revised Fig.3 - line XX, page XX - and the exact times of measurement in Fig.S4), the averaging is made both in time and space and the error bars indicate the standard deviation. The revised related part is inserted in line 13, page 394-396 of the revised document:

*"The overall comparison of the observed and modelled average PSDs is presented in Fig 8. We consider that all the in-situ airborne measurements and the WRF-L numerical outputs satisfy the defined spatiotemporal collocation criteria. Error bars indicate the corresponding standard deviation."*

In Fig. 8b (Figure 9b in the original manuscript), we present an alternative comparison between observations and model volume concentrations, for the selected AER-D samples. First, we calculated the total dust volume concentration by integrating the modelled and observed PSD of each flight segment. Then, we find the relative difference of the total volume concentration with respect to observations for each flight segment expressed in percentage. That relative difference of the total volume concentration is denoted in Fig 9b (using different marker styles) with respect to the specific altitude that the flight segment occurred. The dashed

coloured vertical lines show the corresponding differences (in percentage) spanning from near-surface up to ~4.2 km. The different colours correspond to the different numerical experiments. These differences are calculated as the average of the relative differences of each flight segment. The revised related part is inserted in lines 403-407, page 14 of the revised document:

*"More specifically, we calculate for each model experiment (denoted with different colour), the relative differences (expressed in percentage) of the total dust volume concentration with respect to the in-situ measurements. In addition, the corresponding differences (in percentage terms) that are representative for the altitudes spanning from near-surface up to ~4.2 km are denoted with the vertical coloured dashed thick lines (WRF-L experiments). Those differences are derived by averaging the relative differences of each flight segment."*

**L 14 Why is there a limit of dust particle sizes (0.2 < D < 100 um), in particular in the context of observations?**

We agree with the reviewer that such a statement about the size limits of the observed dust particles here is wrong. Since there is a detailed reference in the Introduction about the sizes of the observed dust particles in the atmosphere, we have omitted the content of the parenthesis from the abstract of the original manuscript. Nevertheless, we should note that both Fennec and AER-D field campaigns provided observations of particles much larger than 100 μm, but AER-D was curtained at 100 microns because few larger particles were detected, resulting in instrument noise at these sizes. The same is true for the Fennec campaign, for particles larger than 500 microns. Based on our reply we have modified the revised manuscript in lines 13-16, page 1:

*"Dust particles larger than 20 μm in diameter have been regularly observed to remain airborne during long-range transport. In this work, we modify the parameterization of the mineral dust cycle in the GOCART-AFWA dust scheme of WRFV4.2.1, to include also such coarse and giant particles, and we further discuss the underlying misrepresented physical mechanisms which hamper the model in reproducing adequately the transport of the coarse and giant mineral particles."*

**L 15 The formulation "extend the parameterization of the mineral dust cycle" is not suitable. The parameterization of the mineral dust cycle (emission, transport [which includes itself several parameterizations], and deposition [again more than one parameterization]) was not extended, but some aspects of it were modified. The same applies for "our parameterization" (L 17).**

We agree with the reviewer that the choice of the verb "extend" is not suitable for describing our methodology, thus we have modified the related part in the revised manuscript in lines 14-16, page 1:

"*In this work, we modify the parameterization of the mineral dust cycle in the GOCART-AFWA dust scheme of WRFV4.2.1, to include also such coarse and giant particles, and we further discuss the underlying misrepresented physical mechanisms which hamper the model in reproducing adequately the transport of the coarse and giant mineral particles.* "

**L 21 - 22 Those additional processes have been proposed in the past, hence this statement is inaccurate. I suggest revising it and stating (after mentioning the sensitivity experiments) that those processes are discussed as candidates to cause such a reduced settling.**

We agree with the reviewer, thus we have revised the related parts of the revised manuscript in the abstract according to the reviewer's suggestion in lines 20-23, page 1

"*The results show that the modelled lifetimes of the coarser particles are shorter than those observed. Several sensitivity runs are performed by reducing artificially the particles' settling velocities in order to compensate underrepresented mechanisms, such as the non-spherical aerodynamics, in the relevant parameterization schemes.*"

**L 24 in the range**

Done

**L 25 UR60 has not yet been introduced**

Done

**L 30 Important to mention that dust only ranks first/second by mass.**

Here we state that dust aerosol ranks first in mass burden and second in emission. This is based on median values from all models that participated in AEROCOM Phase III. Those results, which are close to the results of AEROCOM Phase I (*Huneeus et al., 2011;* Textor et al., 2006), are depicted in Table 3 of Gliß et al., (2021). Based on our answer modified the lines 32-33, page 2 of the revised manuscript

*"Dust is the most prominent contributor to the global aerosol burden, in terms of dry mass, and it ranks second in aerosol emissions (Gliß et al., 2021; Huneeus et al., 2011; Textor et al., 2006)."*

**L 32 Dust can be windblown, but I believe the emissions cannot.**

Absolutely. Corrected.

**L 34 Aren't all regions "spatially limited"? Perhaps use "Spatially more limited".**

Thank you. Corrected.

**L 41 after their wet and dry deposition**

Done

**L 46 I propose "cloud microphysical processes and their evolution" [omit the dissolution part] as I believe the processes do not stop.**

Absolutely. Corrected.

**L 51 Please give a reference for this diameter range. The lower limit seems relatively large to me.**

We would like to thank the reviewer for pointing this out. Of course, there are smaller dust particles in the atmosphere than 0.2 µm. Our statement is

erroneously driven by the range of the model sizes. Thus, we have revised the manuscript by removing that statement.

**L 65-66 gravitational settling**

Done

**L 69 of all cases**

Done

**L 70 Please give a (spatial) reference for "larger distances"**

In Weinzierl et al., 2017a authors estimated that particles with diameters 10-30 μm occur approximately at 2000km further from their sources, than it is expected, based on Stoke's law, under no shear conditions and assuming the initial height of the observations taken. Based on that, we added the spatial reference for larger distances in the manuscript revising the original manuscript in line 72, page 3:

*"Dust particles with diameters of 10 to 30 μm were detected during the SALTRACE campaign in Barbados (Weinzierl et al., 2017a), revealing that they were suspended far away from their sources at about 2000 km more than what would be expected from the Stokes' theory (Weinzierl et al., 2017a)"*

**L 71 Stokes' theory is on settling, not on gravity.**

Absolutely. Corrected.

**L 112 The modified model version considers dust up to 100 microns, but airborne dust particles can also be larger, hence "the entire size range" is exaggerated.**

Absolutely. We rephrase the statement in line 116, page 4 of the revised manuscript.

*"in **step 2**, we define five size ranges (five model size bins) for the transported PSD covering dust particle sizes spanning from 0.2 μm to 100 μm (Sect. 2.1.1);"*

**L 131 Please add [in the default GOCART-AFWA] dust emission scheme [of (in) WRF]**

Done.

**L 152 Please introduce variables directly to Eqs. 3 and 4.**

Done.

**L 160 The Cunningham correction is missing in Eq. 5**

We would like to thank the reviewer for noticing these errors. In the revised manuscript, we have corrected equations and based RC2 reviewer suggestions we reconstruct the Section 2.1.2 to make it less lengthy. In the revised text Cunningham correction is introduced in Eq.4 and is given by Eq.5. The revised Section 2.1.3 (see lines 156-214, pages 6-8) is shown below:

[revised manuscript text omitted]

$(12)"$

**L 172/176 I know the drag coefficient equation for the Stokes regime as C_D = 24 / Re with Re = U D / nu with nu = mu / rho. Is there any reason I am missing why the formulation shown here is different and contains the factor 2 in Re rather than C_D? (The result is the same.)**

We can understand the reviewer's confusion. All equations are written in such a way to be a function of particle diameter.

**L 178/192 The Kelvin scale has no degree symbol.**

Thank you for pointing this out. It is corrected.

**L 182 I don't think Equation 7 is meant here. Equation 4 maybe?**

Thank you. It is corrected.

**L 184 delete become**

Done.

**L 193 Eq. 13?**

Thank you. Corrected.

**L 200 remove parenthesis around first reference**

Done.

**L 208 Re < 1?**

The limit Re<0.1 is correct. There is a more detailed explanation for the reasons behind that on Mallios et. al 2020. However, we have revised the revised document to be clearer in lines 207-214, pages 7-8:

"In the default WRF code the slip correction is applied unconditionally for all the Re values, probably without affecting the solution significantly due to the small particle sizes ($D_{eff} < 20$ μm). However, in our work required a condition is required for applying the slip correction only in the Stokes' regime (e.g. Re < 0.1, Mallios et. al, 2020). Hence, we apply the bisection method to calculate the terminal velocity for each model size bin using the revised drag coefficient and, at first, ignoring the slip correction. When the solution lies in the Stokes' regime (e.g. Re < 0.1), we recalculate the settling velocity using the corrected drag coefficient

$C'_{D,slip} = \frac{C'_D}{C'_{cun}}$ , where $C'_{cun} = C_{cun}(\lambda')$ with $\lambda'$ the mean free path obtained by (Jennings, 1988):

$$\lambda' = \sqrt{\frac{\pi}{8}} \cdot \frac{\frac{\mu}{0.4987445}}{\sqrt{P\rho_{air}}} ,$$
(12)"

**L 218 Why did the authors choose to include so much ocean in their domain while omitting east N African dust sources? This seems not an ideal choice to me.**

We made the choice to use this domain because the airborne in-situ observations of the PSD have been acquired in the surrounding area of Cape Verde, so we decided to have this location very close to the centre of our computational domain. Moreover, the main sources affecting this area of interest during the boreal summer (August 2015), are encompassed within the simulation domain. Dust sources in East Africa have negligible effect on dust concentrations over Cape Verde, so we decided to omit them. Ideally the model domain can be increased towards the East and the North,

but the computational cost is large compared to any small anticipated benefits.

**L 225-226 The authors state that "scaling of the dust source strength is chosen to best match the modeled DOD with the AERONET measurements". I would like to know more about this. What scaling do you refer to? Is this a universal scaling/tuning factor or a map scaling? Did you modify the Ginoux/GOCART erodibility function typically used in WRF or is a different scaling used? How has the modeled dust been compared with the observations to infer any kind of scaling? Please give more detail as this is an important aspect of the modeled dust fields.**

We adjust the empirical constant in the equation of horizontal saltation fluxes emission (Cmb in Eq.10 in LeGrand et al., 2019) in order to have the best statistical agreement between AERONET filtered AOD and model DOD. Based on our reply we have include additional information in line 229-235, page 8:

"We scale the dust source strength, by tuning the empirical proportionality constant in the horizontal saltation flux equation (in eq. 10 in LeGrand et al., (2019)) in order to obtain the best match between the modeled DOD and the AERONET AOD (RMSE=0.34, bias=-0.07) acquired at 8 desert stations: Banizoumbou, Dakar, El_Farafra, Medenine- IRA, Oujda, Tizi_Ouzou, Tunis_Carthage, Ben_Salem). Note that we take into account only AERONET records when AODs are higher than 0.2 (Version 3.0, Level 1.5, Giles et al., 2019; Sinyuk et al., 2020) and the Angstrom exponent is lower than 0.75. The tuning constant is equal to 3 and is applied throughout the model domain."

**L 228 A minimum DOD of 0.75 seems very high to me, even close to dust sources.**

We would like to thank the reviewer for noticing that. Actually, the filtering of AERONET dust AOD-like is made by defining a lower (0.2) and an upper (0.75) threshold on AOD and Angstrom, respectively. The necessary clarifications have been incorporated in the revised manuscript in line 232 , page 8:

"Note that we take into account only AERONET records when AODs are higher than 0.2 (Version 3.0, Level 1.5, Giles et al., 2019; Sinyuk et al., 2020) and the Angstrom exponent is lower than 0.75."

**L 233 by up to 80 % with a step size…**

Done.

**L 234 "sensitivity experiment" instead of "artificial tuning"**

Done.

**L 235 Please revise "falling into the atmosphere"**

We agree with the reviewer that this statement needs revision, thus we have modified the original manuscript, based on this comment and the next one in lines 244, page 9 of the revised document:

*"A series of additional sensitivity runs has been performed aiming to resemble possible mechanisms (misrepresented or even absent in the model) counteracting gravitational settling towards reducing the differences between the CONTROL run calculations and the in-situ observations (shown in Sect. 3.4). To be more specific, we gradually reduced (with an incremental step of 20%) the settling velocity by up to 80%, with the corresponding runs named as URx (x corresponds to the reduction in percentage terms). Under such theoretical conditions, it is expected that the giant dust particles will be suspended for longer periods and that they will be transported at larger distances than the current state-of-the-art models simulate, failing to reproduce what is observed in the real world. Based on these sensitivity experiments, we defined a constant (by percentage) relevant reduction of the particles' settling, which in its absolute value varies with size. Therefore, it is more similar to the effects that are related to aerodynamic forces due to the non-spherical shape and the orientation of the suspended dust particles (Ginoux, 2003b; Loth, 2008; Zastawny et al., 2012; Shao et al., 2017; Sanjeevi et al., 2018; Mallios et al., 2020)."*

**L 236 "all real forces" is exaggeration. Gravitation and drag forces are real and**

Absolutely. We revised the related statement in line 250, page 9 of the revised manuscript:

*"Based on these sensitivity experiments, we defined a constant (by percentage) relevant reduction of the particles' settling, which in its absolute value varies with size. Therefore, it is more similar to the effects that are related to aerodynamic forces due to the non-spherical shape and the orientation of the suspended dust particles (Ginoux, 2003b; Loth, 2008; Zastawny et al., 2012; Shao et al., 2017; Sanjeevi et al., 2018; Mallios et al., 2020)."*

**L 240 What "fine resolution" are your referring to here? I would not consider 15 km a particularly fine resolution. Also, Table 3 does not contain any experiments on resolution (L 248).**

The resolution applied here is adequate for the scale of phenomena we want to study. By "fine" resolution we wanted to denote that we have a finer resolution than global datasets (e.g. 0.5deg GFS) which will fail to reproduce the appropriate weather fields. However the reviewer is correct that this can be misleading so we made changes to the text to be clearer in line 235, page 8.

*"The resolution applied in this study (15km grid spacing) is adequate for the scale of phenomena we want to study, improves the representation of topography and increases the accuracy of the reproduced weather and dust fields, compared to coarser resolution, such as used in global datasets (e.g. 0.5 deg GFS) (Cowie et al., 2015; Basart et al., 2016; Roberts et al., 2017; Solomos et al., 2018)."*

**L 243 Dunes are no meteorological condition.**

Absolutely. Based on other comments we have removed this part.

**L 270 The explanation is hard to understand, please revise it if possible. How did you handle missing values in the observations for the model comparison?**

Obviously, the description needs improvement. In the model, the DOD is computed in each grid model box and its instantaneous value is provided every one hour. The DOD value from Aqua satellite is acquired from the ModIs Dust AeroSol (MIDAS) DOD product, based on the following spatiotemporal collocation procedure: First, we reproject the model DODs on an equal lat-long grid at 0.4° x 0.4° spatial spacing. We should note that the model DOD field has no spatial gaps and is provided instantaneously for every hour. The MIDAS DOD is available in swath level (5-minute segments, viewing width of 2330 km) along the MODIS-Aqua polar orbit. Then, the two closest WRF outputs to the Aqua satellite overpass time are used to calculate a weighted-average WRF-DOD, by taking into account the

temporal departure between forecast and overpass times, only for the WRF grid cell that coincides with the observations. Please note that we have removed the corresponding part related to b920 flight, based on RC3 comment. Please note that based on next comment suggestion about the discussion of Fig.7 of the original manuscript, the related part has been removed.

**L 288 I suggest mentioning here again how the FENNEC PSD has been used. This will be as brief as mentioning that it is explained elsewhere (you can keep the reference toy Sec. 2.1.1).**

We agree with the reviewer's and other reviewers' suggestions and we have revised the whole 2.2.1 accordingly (see lines 278-306, pages 10-11):

*"During the FENNEC field campaign in 2011 (Ryder et al., 2013b, 2013a) and the AER-D field campaign in 2015 (Ryder et al., 2018, 2019), airborne in situ observations were collected with the FAAM BAE research aircraft.*

*In this study we use size distributions from the FENNEC field campaign, aquired during aircraft profiles over the Sahara (Mauritania and Mali), as described in Ryder et al. (2013a). We select size distributions from "freshly uplifted dust" cases, when dust particles are in the atmosphere for less than 12 h. Additionally, from these profiles we use data from the lowest available altitude, centered at 1km, covering altitudes between 0.75 to 1.25km. The derived PSD is depicted in Fig.2(a), hereafter referred to as the "observed FENNEC-PSD". Error bars in Fig.2(a) indicate the standard deviation of the observed values across the profiles and altitudes we used. The instrumentation for those measurements was the Passive Cavity Aerosol Spectrometer Probe (PCASP, 0.13-3.5 μm), the Cloud Droplet Probe (CDP, 2.9-44.6 μm), using light scattering measurements and assuming a refractive index (RI) of 1.53-0.001i (which is constant with particle size), spherical shape for the particles, and using Mie calculations to convert from optical to geometric diameter, as well as the Cloud Imaging Probe (CIP15, 37.5-300 μm)). The instruments and data processing are described in Ryder et al. (2013a). The midpoint size bin diameters do not overlap, though there is some overlap in bin edges between the instruments. A fit on the observations is provided in Figure 2a (the "fitted FENNEC-PSD" with solid red line), which is used in the parameterization of the emitted dust, as described in Section 2.1.1, to modify the GOCART-AFWA dust scheme in WRF.*

*We also use PSD observations during horizontal flight legs at a constant height (referred either as RUNs or flight segments) over the Atlantic Ocean during AER-D. We use measurements taken with PCASP (D =0.12-3.02 μm) for fine dust particles. For the coarse and giant mode of dust we used measurements from CDP (D=3.4-20 μm, although CDP measurements availability extends up to 95.5 μm as it is explained below) and the two-dimension Stereo probe (2DS, D = 10–100 μm -although the instrument measures up to 1280 μm few particles*

*larger than 100μm were detected). For the light scattering techniques of PCASP and CDP, a RI = 1.53-0.001i is assumed for the conversion of the optical to geometric diameter (as in FENNEC 2011 campaign). CDP observations extend up to the size of 95.5 μm, thus data from CDP and 2DS partly overlap in their size range. Since 2DS observations are more reliable in the overlapping size range, we used the CDP observations for particles with sizes up to 20 μm. Also, 2DS-XY observations are preferred over the 2DS-CC, since they better represent the non-spherical particles. A more detailed description of the in-situ instruments and the corresponding processing of the data acquired during the AER-D campaign is included in Ryder et al., (2018). The error bars represent the total (random and systematic) measurement error due to the counting error, the discretization error, the uncertainties in the sample area and the uncertainties in the bin size due to Mie singularites (Ryder et al., 2018). All PSD measurements are at ambient atmospheric conditions. The locations of the flights of AER-D used in this study are depicted in Fig.3."*

**L 338/Fig. 5 Are the modelled PSDs for a particular time step or averaged?**

It is at the hourly model output at 15 UTC, which contains the model values for the corresponding time step. The information is included in lines 334-336, page 12 of the revised manuscript:

*"In Fig. 5 we present how the PSD varies with height above an emission point (latitude=24.9º and longitude=9.2º) in Mali, on 11/08/2015 at 14UTC. The model PSDs are only from that grid model box interpolated at 1, 2, and 3 km height and for the particular timestep (11/08/2015 at 14UTC) ."*

**L 360 I suggest showing deposition rates for bin 5 to see whether all particles have settled already over land.**

We have added in Fig. 6g a map for the time-averaged (from 5-25/8/2015) gravitational deposition rate of particles in bin 5. The map shows that the major mass of particles of size 5 is deposited not further than the African coasts. Almost all dust is deposited not further than the parallel of 20º W. The revise Fig 6 is inserted in lines 923-926, page 35 of the revised manuscript:

"

[Figure]

*Figure 6: The dust load provided by the model, averaged for the whole simulation period, for (a) bin 1, (b) bin 2, (c) bin 3, (d) bin 4, (e) bin 5, and (f) the whole range of the PSD. The dust load is in g/m2. (g) The gravitational deposition rate for bin 5.*
"

**L 363 – 375 The discussion about Flight b920 in the context of Fig. 7 is a bit confusing as Fig. 7 does not contain the PSD measured during the flight (but only the displaced dust plume). Why don't you show the PSD from b920 to provide a basis for the discussion?**

We agree with the reviewer that the discussion about flight b920 is a bit confusing, thus we excluded it for the manuscript.

**L 393 The relative difference shown in Fig. 9b does not seem to vary systematically with height for bin 5.  Shouldn't this be expected?**

We would like to clarify that Figure 8b (Figure 9b in the original manuscript) shows the relative difference of the Total Volume concentration (sum of the concentration for the five model bins) which is contained in the PSD of each flight segment (different markers) versus the altitude of each flight segment. With different colours are the results from the different sensitivity tests and not for the different model size bins.

**L 397 What kind of average is the "mean extinction coefficient"?**

The LIVAS mean extinction coefficient is obtained by averaging all the LIVAS profiles per CALIPSO nighttime overpasses between 25.5°W to 12.5°E and 11.5°N to 34.5°N.

For the same overpasses, we obtained the model profiles collocated as described in Section 2.1.4. We thank the reviewer for noticing that the description needs improvements, so we revise the document accordingly in page 14 and lines 416-425 (of the revised document):

*"Figure 9(a) shows the profile of the mean extinction coefficient at 532 nm, provided by the LIVAS pure-dust product (black line), and the profile of the mean extinction coefficients at 550 nm, provided by the CONTROL, UR20, UR40, UR60, and UR80 experiments. The orange area indicates the standard deviation of the LIVAS profiles. Figure9(b) depicts the mean absolute model bias with respect to LIVAS profiles for the different simulations and the vertical dashed lines show the corresponding bias averaged over different altitudes. The mean LIVAS profile is provided by averaging the night-time profiles over the region bounded by between 25.5°W to 12.5°E and 11.5°N to 35.5°N and from dates spanning from 5 to 25 August 2015. This area includes the main dust sources that affected the vicinity of Cape Verde (Ryder et al., 2018) and the region of the dust outflow over the Ocean, as well. The nightime profiles excel in accuracy over the daytime ones, due to the lower signal-to-noise ratio (SNR) during the night. The model profiles are collocated in space and time with the LIVAS profiles, as described in Sect. 2.1.4 and the model extinction coefficient is provided with the Eq.13."*

**L 399-400 It has been discussed before how a few dust plumes were displaced, hence I do not agree with this general affirmation of simulation quality.**

We agree with the reviewer that there are departures in the vertical distribution of dust, thus we remove the affirmation.

**L 402 How can these mean (?) profiles be related to the night-time boundary layer? Was any more detailed analysis performed?**

In the framework of the present study, we implement LIVAS (LIdar climatology of Vertical Aerosol Structure for space-based lidar simulation studies), a 3-D multi-wavelength global aerosol and cloud optical database developed in the framework of the European Space Agency (ESA) activities, towards providing support for future satellite-based lidar missions (Amiridis et al, 2015). The LIVAS database provides vertical profiles of aerosol optical properties, including among others L2 QA profiles of extinction coefficient at 532 nm, not only for the total aerosol load, but also for the pure-dust component (Amiridis et al., 2013; Marinou et al., 2017; Proestakis et al., 2018), through implementation of an EARLINET (Pappalardo et al., 2014) established methodology (Tesche et al., 2009). However, the original ESA-LIVAS database which is implemented to address the scientific questions of the present study, does not provide the PBL information. Moreover, although MERRA-2 is extensively implemented in the framework of CALIOP algorithms, CALIPSO L2 Aerosol/Cloud Profiles 5 km do not provide information on PBL at per-CALIPSO orbital level. Thus, neither CALIPSO nor the ESA-LIVAS database, the observational lidar-based satellite products and datasets extensively analysed and implemented as references towards the evaluation of the model-based outputs in the framework of the study, provide input on the PBL, to be used here. Estimation is based on the shape of the climatological vertical structure of the mean extinction coefficient profile at 532 nm, over RoI and for the period of interest, and additionally experimental (i.e., Ansmann et al., 2011; Weinzierl et al., 2016) and

climatological (Marinou et al., 2017) studies over the domain. However, we would like to thank the reviewer, for it is clear that the section of the manuscript needed improvement. Thus the sentence is modified accordingly in lines 426-428, page 14 of the revised manuscript:

"*The intercompared profiles are in a good agreement, with the simulations falling well-within the variability of the dust observations, although discrepancies are also present, especially close to the dust sources (Fig.9(b) – region I), and within the upper free Troposphere (Fig. 9(b) – region III).*"

**L 403-407 This discussion sounds like the observations are the main cause for model-observations discrepancies. I understand that this discussion is done to provide a justification why only Region II has been assessed. I suggest to revise the wording to avoid misinterpretation.**

The reviewer is right on his guess, and we would like to thank the reviewer for the comment. Following the comment, and in order to avoid possible misinterpretations, the following section from the manuscript is modified in line 428-430, page 14 of the revised manuscript:

"*The assessment of the different model experiments against the ESA-LIVAS pure-dust product is performed in the region between 1.5 km and 6.4 km a.m.s.l. (Fig. 9 – region II), to avoid possible biases propagating into the analysis (i.e., complex topography and surface returns-region I, SNR and tenuous aerosol layers – region II)*".

**L 418 "acknowledged" is not the right word here, neither "transport code".**

Absolutely, we proceed revising the related text with more appropriate wording in line 440, page 15 of the revised document:

"*In this study we extend the particle size range which is applied in the transport parameterization in GOCART-AFWA dust scheme of WRF, to include particles with diameters up to 100 μm..*"

**L 440 "(two times the particle major semi-axis)" seems out of place.**

We have removed this part revisiting the section of Discussion base on the reviewer's and other reviewers' suggestions.

**Discussion: I believe that much of the discussion around the different processes that might affect particle transport should go into the introduction. Only the discussion around the percentages in reduced settling these processes might account for should remain in the discussion.**

We would like to thank the reviewer for the suggestion, we have put a lot of effort in revising the Discussion Section, hoping that it is better and more understood. Bellow is the part of the Discussion (see lines 436-503, pages 15-17 of the revised manuscript):

[revised manuscript text omitted]

**L 482 losses instead of loses**

Done.

---

## Author Response (AR2)

Dear Editor,

Thank you for agreeing to consider a revision of our manuscript "Modelling coarse and giant desert dust particles". We modified and revised the manuscript to address the reviewers' comments as well as to clarify points that they found confusing or unclear.

We would like to thank the two anonymous reviewers for their helpful comments and suggestions, and many thanks to you for your time and efforts with this revision. In line with the comments and suggestions, we revised the manuscript and made the requested additions and changes. Below are all the comments (in bold) followed by the replies. The parts that are in italic are corrections that are included in the revised version of the paper:

Sincerely, Eleni Drakaki

**Anonymous Referee #1**

This study investigates the incorporation of coarse and giant desert dust particles (with diameter greater than 20  $\mu$ m) in the WRF model, together with the GOCART aerosol model and the AFWA dust emission scheme. The authors implemented a number of extensions to the original model. More specifically, they used a prescribed dust particle size distribution for emitted dust particles at the source based on in situ measurements from the FENNEC campaign and employed 5 size bins with diameters up to 100  $\mu$ m (corresponding to giant particles). Moreover, they implemented an updated drag coefficient that applies to the above bins and is representative of high values of Re number. The simulations were performed from 29 July to 25 August 2015. The model output were validated against various observational datasets.

The article is well written and promotes the research in the modelling of the desert dust. The use of English is excellent

**and the conclusions are supported by the results. It is suggested to accept this article for publication after some minor corrections are performed.**

The recognition of our work from the reviewer is much appreciated. We would like to thank him/her for taking the necessary time and effort to review our manuscript. We sincerely appreciate all your valuable comments and suggestions. The manuscript has been revised considering all the suggestions raised by the reviewer.

**Suggested corrections:**

Section 2.1.3: please include a) whether the vertical levels (line 220) were defined by WRF or by the authors (providing how you chose them in the latter case), b) which UTC time was chosen for the original initialization/each reinitialization (line 221), c) some more detailed information about the model results that you used from each 84 hour run (i.e. whether you removed the first 12 hours of each run due to model spin-up and utilized the rest; line 221), d) the topography and land-use datasets, e) whether the seasurface temperatures were updated from GFS-FNL analyses every 72 hours at the initial time of each run or every 6 hours together with the lateral boundary conditions.

We agree with this comment and we have incorporated the reviewer's suggestion throughout Section 2.1.3, explaining that the specific heights of the vertical levels are defined by the model. The sea surface temperatures in the model acquired by the NCEP daily SST analysis (RTG\_SST\_HR) are updated every six hours along with the lateral boundary conditions. Each 84-hour run was initialized at 12 UTC and the first 12 hours were removed accounting for the model spin-up. Topography is interpolated from the 30s Global Multi-resolution Terrain Elevation Data 2010 (GMTED2010, Danielson and

Gesch, (2011)). We use land-data based on Moderate-resolution Imaging Spectroradiometer (MODIS) observational data modified by the University of Boston (Gilliam and Pleim, 2010). Hence based on our reply we modified the Sect 2.1.3 of the original manuscript (line 216, page 8):

"Using the WRF-L code, we first run the CONTROL experiment. Our simulation period coincides with the AER-D experimental campaign (29/7 - 25/8/2015) for a domain bounded between the 1.42-N and 39.99-N parallels and stretching between the 30.87-W and 46.87-E meridians (Fig. 3). The simulation area encompasses the major Saharan sources also including the downwind areas in the eastern Tropical Atlantic. We use an equal-distance grid with a spatial grid spacing of 15 km x 15 km consisting of 550 × 300 points whereas in vertical, 70 vertical sigma pressure levels up to 50 hPa are utilized. The simulation period consists of nine 84-hour forecast runs, which are initialized at 12 UTC, using the 6hour Global Forecast System Final Analysis (GFS - FNL) reanalysis product, available at a 0.25-x0.25spatial resolution. The sea surface temperatures, acquired by the NCEP daily global SST analysis (RTG\_SST\_HR), are updated every six hours along with the lateral boundary conditions. From each 84hour cycle, the first 12 hours are discarded due to model spin up. Likewise, the first week of the simulation served as a spin-up run for the accumulation of the background dust loading and it is excluded from the analysis."

Line 369-373: Have you validated the simulated upper air wind field, e.g. using ERA5? Western Africa is characterized by a complex wind regime. There is a large area with pink colors (i.e. dust) in area B of Figure 7f. Therefore, the dust errors may be also due to erroneous wind field.

The reviewer raises an important issue regarding how a possible wind speed bias can affect the emission. Menut, (2008) quantified the impact of the meteorological data forcing (using either NCEP or ECMWF as initial/boundary conditions) above Sahara sources and reported that the difference between the two emission fluxes can reach a factor of 3. Moreover, they noted that this difference is not systematic and no conclusion was made of which dataset overperforms. Following the reviewer's suggestion, we performed a validation analysis of the WRF-L upper wind fields (i.e., at 300 and

500 hPa) versus ERA5, both reprojected at a common grid (0.25 x 0.25 spatial resolution). The obtained results for the two pressure levels, on 5th August 2015 at 00 UTC, are illustrated in the Figure below. It is evident that the two models produce similar meteorological patterns with deviations only on the wind speeds. Focusing on the latitudinal band (10-25°N) where the Saharan dust is transported over the Tropical Atlantic Ocean, mainly positive WRF-ERA5 declinations are recorded over the W. Sahara while the opposite is revealed over the outflow regions. This differences in the two models above land are almost consistent throughout the whole simulation. In terms of magnitude lie mostly in the range of 2-8 m/s (in absolute terms) at 500 hPa and they are slightly higher at 300 hPa.